# Structural polymorphism of *α*-synuclein fibrils alters the pathway of Hsc70-mediated disaggregation

Svenja Jäger[1], Jessica Tittelmeier [1,2], Thi Lieu Dang[1], Tracy Bellande[3], Virginie Redeker [3], Alexander K Buell [4], Janosch Hennig [5,6], Ronald Melki [3✉], Carmen Nussbaum-Krammer [1,2✉], Bernd Bukau [1✉] & Anne S Wentink [1,7✉]

## Abstract

**Pathological aggregation of α-synuclein into amyloid fibrils is a hallmark of synucleinopathies, including Parkinson's disease. Despite this commonality, synucleinopathies display divergent disease phenotypes that have been attributed to disease-specific three-dimensional structures of α-synuclein fibrils, each with unique toxic gain-of-function profiles. The Hsc70 chaperone is remarkable in its ability to disassemble pre-existing amyloid fibrils of different proteins in an ATP and co-chaperone-dependent manner. We find, however, using six well-defined conformational polymorphs of α-synuclein fibrils, that the activity of the Hsc70 disaggregase machinery is sensitive to differences in the amyloid conformation, confirming that fibril polymorphism directly affects interactions with the proteostasis network. Amyloid conformation influences not only how efficiently fibrils are cleared by the Hsc70 machinery but also the balance between depolymerization and fragmentation during disaggregation. We further show that, in vitro, the active processing of fibrils by the Hsc70 machinery inadvertently produces seeding competent species that further promote protein aggregation. Amyloid conformation thus is an important feature that can tilt the balance between beneficial or detrimental protein quality control activities in a disease-context.**

**Keywords** Alpha-synuclein; Polymorphism; Disaggregation; Hsp70; Molecular chaperones
**Subject Categories** Neuroscience; Post-translational Modifications & Proteolysis

## Introduction

A hallmark of neurodegenerative diseases is the accumulation of amyloid-type aggregate deposits of signature proteins in patient brains. In synucleinopathies such as Parkinson's disease (PD),

dementia with Lewy bodies (DLB), or multiple system atrophy (MSA), it is the intrinsically disordered protein α-synuclein (α-syn) that aggregates into such amyloid fibrils (Spillantini et al, 1997, 1998; Wakabayashi et al, 1998). Despite this common feature, the distribution and progression of pathological aggregate deposits differ strongly between diseases (Goedert, 2001; Barker and Williams-Gray, 2016; Holec and Woerman, 2021). It has been suggested that conformational differences in the amyloid structure adopted by α-syn, so-called polymorphism, underlie the observed differences between synucleinopathies. This hypothesis is based on the observation that the amyloid structure found in MSA patients is distinct from that of Lewy body diseases, DLB and PD (Schweighauser et al, 2020; Yang et al, 2022). The resulting differences in surface-exposed residues can alter the interactions with cellular factors, leading to different modes and degrees of toxicity and spreading (Peng et al, 2018; Shahnawaz et al, 2020; Bousset et al, 2013; Van der Perren et al, 2020; Barker and Williams-Gray, 2016; Peelaerts et al, 2015; Hoppe et al, 2021).

As guardians of cellular proteostasis, molecular chaperones play an important role in preventing and reversing protein aggregation (Wentink et al, 2019; De Mattos et al, 2020; Hipp et al, 2019). The human constitutive 70 kDa heat shock protein, Hsc70, and its co-chaperones, the J-domain protein DnaJB1 and the Hsp110 nucleotide exchange factor, Apg2, have been shown to efficiently resolubilise preformed amyloid fibrils of α-syn, the exon 1 of Huntingtin, and different isoforms of tau (Scior et al, 2018; Duennwald et al, 2012; Nachman et al, 2020; Gao et al, 2015; Wentink and Rosenzweig, 2023). In this process, DnaJB1 identifies the amyloid fibrils as a target via multivalent interactions with repeated low-affinity binding sites in the flexible tails protruding from the amyloid fibrils and initiates the recruitment of Hsc70 (Wentink et al, 2020; Gao et al, 2015; Ayala Mariscal et al, 2022). Substrate recognition by DnaJB1 is important to efficiently cluster several Hsc70s on the fibril in functional cooperation with Apg2, generating an entropic pulling force that destabilizes the fibril, leading to disaggregation (Faust et al, 2020; Wentink et al, 2020; Beton et al, 2022). An

[1]Center for Molecular Biology of Heidelberg University (ZMBH), DKFZ-ZMBH Alliance, Heidelberg, Germany. [2]Chair of Neuroanatomy, Institute of Anatomy, Faculty of Medicine, Ludwig-Maximilians-University of Munich (LMU), Munich, Germany. [3]Institute Francois Jacob (MIRCen), CEA, and Laboratory of Neurodegenerative Diseases, CNRS, Fontenay-Aux-Roses, France. [4]Department of Biotechnology and Biomedicine, Technical University of Denmark (DTU), Kgs, Lyngby, Denmark. [5]Chair of Biochemistry IV - Biophysical Chemistry, University of Bayreuth, Bayreuth, Germany. [6]Molecular Systems Biology Unit, European Molecular Biology Laboratory, Heidelberg, Germany. [7]Leiden Institute of Chemistry (LIC), Leiden University, Leiden, Netherlands. ✉E-mail: ronald.melki@cnrs.fr; carmen.nussbaum@med.uni-muenchen.de; b.bukau@zmbh.uni-heidelberg.de; a.s.wentink@lic.leidenuniv.nl

important binding site for DnaJB1 in α-syn is located in the C-terminal tail, while Hsc70 binding sites are present primarily in the N-terminus of the α-syn sequence (Redeker et al, 2012; Wentink et al, 2020; Nury et al, 2015; Burmann et al, 2020). In the context of different polymorph structures, these distinct binding sites might be buried or sterically occluded. This raises the question whether different fibrillar α-syn polymorphs show differences in susceptibility to disaggregation by the human chaperone machinery.

An Hsc70-mediated disaggregation mechanism based primarily on depolymerization, by monomer extraction from fibril ends, has been proposed (Beton et al, 2022; Franco et al, 2021; Schneider et al, 2021). However, the formation of smaller fibrillar fragments during the disaggregation of α-syn fibrils has also been documented (Beton et al, 2022; Gao et al, 2015). This mechanistic distinction is important as the extraction and accumulation of α-syn monomers is likely beneficial, whereas shorter fragments or oligomeric species produced as a result of fragmentation could constitute novel seeds for further amyloid aggregation and favor the propagation of α-syn pathology in a prion-like fashion (Marrero-Winkens et al, 2020; Tittelmeier et al, 2020; Woerman and Bartz, 2024). In vivo studies further highlight this ambiguity in the role of the Hsc70 chaperone machinery in synucleinopathy progression: both the overexpression and knockdown of the Hsp110 co-chaperone have been reported to protect against pathological α-syn aggregation and associated toxicity in animal models (Taguchi et al, 2019; Tittelmeier et al, 2022). Knockdown of the DnaJB1 co-chaperone in reporter cell lines similarly reduced the seeding capacity of some but not all amyloid conformations of α-syn (Tittelmeier et al, 2022). Variation in chaperone-mediated disaggregation kinetics and the resulting products among amyloid polymorphs may therefore contribute to the differences that are observed in disease progression and phenotypes in synucleinopathies (McCann et al, 2014; Hoppe et al, 2021; Tittelmeier et al, 2020).

In this study, we use in vitro aggregated fibrillar polymorphs of human wild-type (WT) α-syn to investigate how the Hsc70 disaggregase accommodates differences in amyloid structure, independent of amino acid sequence differences. These polymorphs are homogenous populations of structurally well-defined α-syn fibrils with distinct features such as a rigid or flexible α-syn N-termini and differences in the periodicity of twist, fibril length, width or stability (Gath et al, 2014; Bousset et al, 2013; Makky et al, 2016; Landureau et al, 2021). Such in vitro aggregated polymorphs are therefore a suitable model to experimentally explore how the disaggregation activity of the human Hsc70 chaperone system is affected by differences in amyloid structures on a biochemical level. We find that fibril conformation alters the ability of the Hsc70 chaperone machinery to recognize the substrate and consequently disaggregate α-syn amyloid fibrils. Polymorphs that were more resistant to chaperone action showed greater accumulation of fibril fragments/oligomeric species during disaggregation, which are highly seeding competent. The presence of a specific amyloid conformation can thus fundamentally alter the ability of the cell's quality control machinery to respond to this threat, which may have profound implications for the onset and progression of disease.

# Results

## α-syn fibrillar polymorphs show different susceptibility to disaggregation by the human Hsc70 chaperone machinery

Recombinant monomeric α-syn protein was aggregated under various buffer conditions (Fig. 1A and "Methods"), resulting in five homogenous, structurally distinct amyloid polymorphs of full-length WT α-syn (FM, Ri, F65, F91, and XG) and one C-terminally truncated mutant (F110) (Fig. 1A and Appendix Fig. S1A) (Gath et al, 2014; Gao et al, 2015; Bousset et al, 2013; Makky et al, 2016; Landureau et al, 2021). To test the susceptibility of the polymorphs to chaperone action, the preformed α-syn fibrils were incubated with the Hsc70-based disaggregase machinery, composed of the constitutively expressed Hsp70 family member Hsc70 (HSPA8), and two essential co-chaperones DnaJB1 and Apg2 (HSPA4/HSPH2). After 16 h, the remaining insoluble fibrils were separated from re-solubilized α-syn by centrifugation (Fig. 1B,C). The sedimentation assay shows that the XG polymorph was efficiently disaggregated by the Hsc70 machinery in an ATP-dependent manner, with 40% of the total α-syn protein recovered in the soluble fraction. For the F65, FM, and F91 polymorphs, 20–30% of α-syn protein was found in the supernatant, while Ri and F110 showed no release of soluble α-syn protein over the 16 h incubation period.

We next compared the kinetics of disaggregation by time-resolved Thioflavin T (ThT)-based disaggregation assay (Fig. 1D,E). ThT is a fluorescent dye that shows enhanced fluorescence intensity upon interaction with β-sheet-rich structures such as amyloid fibrils and can therefore be used to track the dissolution of amyloid fibrils (Biancalana and Koide, 2010). Since ThT shows differential reactivity to the polymorphs (Appendix Fig. S1B), the fluorescence intensity during disaggregation was expressed as a ratio of non-disaggregated fibril intensity (−ATP) to facilitate comparison between the polymorphs. Based on the ThT intensity, both XG and F65 polymorphs were efficiently disaggregated by the Hsc70 machinery within 16 h, with ThT intensities reduced by 55% and 65%, respectively. FM and F91 fibrils showed only moderate disaggregation, with a 15–25% reduction in ThT fluorescence. The reaction kinetics of the disaggregation of the F65 fibrils were noticeably faster in the first hour (1.5-fold) than that of the XG polymorph, despite reaching a similar endpoint (Fig. 1D). The kinetics and endpoint of disaggregation of the FM and F91 polymorphs were comparable, and slower than both XG and F65. In contrast, for ribbons (Ri) and the C-terminal truncated F110 α-syn polymorphs the ThT-signal remained constant over 16 h, indicating resistance to disaggregation by the human Hsc70 chaperone machinery.

Overall, the six α-syn polymorphs can thus broadly be divided into three categories: efficiently disaggregated (F65, XG), moderately disaggregated (FM, F91), and resistant to disaggregation by the Hsc70 chaperone machinery (Ri, F110). Since all fibrillar polymorphs (except for F110) are composed of WT α-syn, the specific conformation of the fibrillar substrate thus directly affects the efficiency of chaperone-mediated disaggregation.

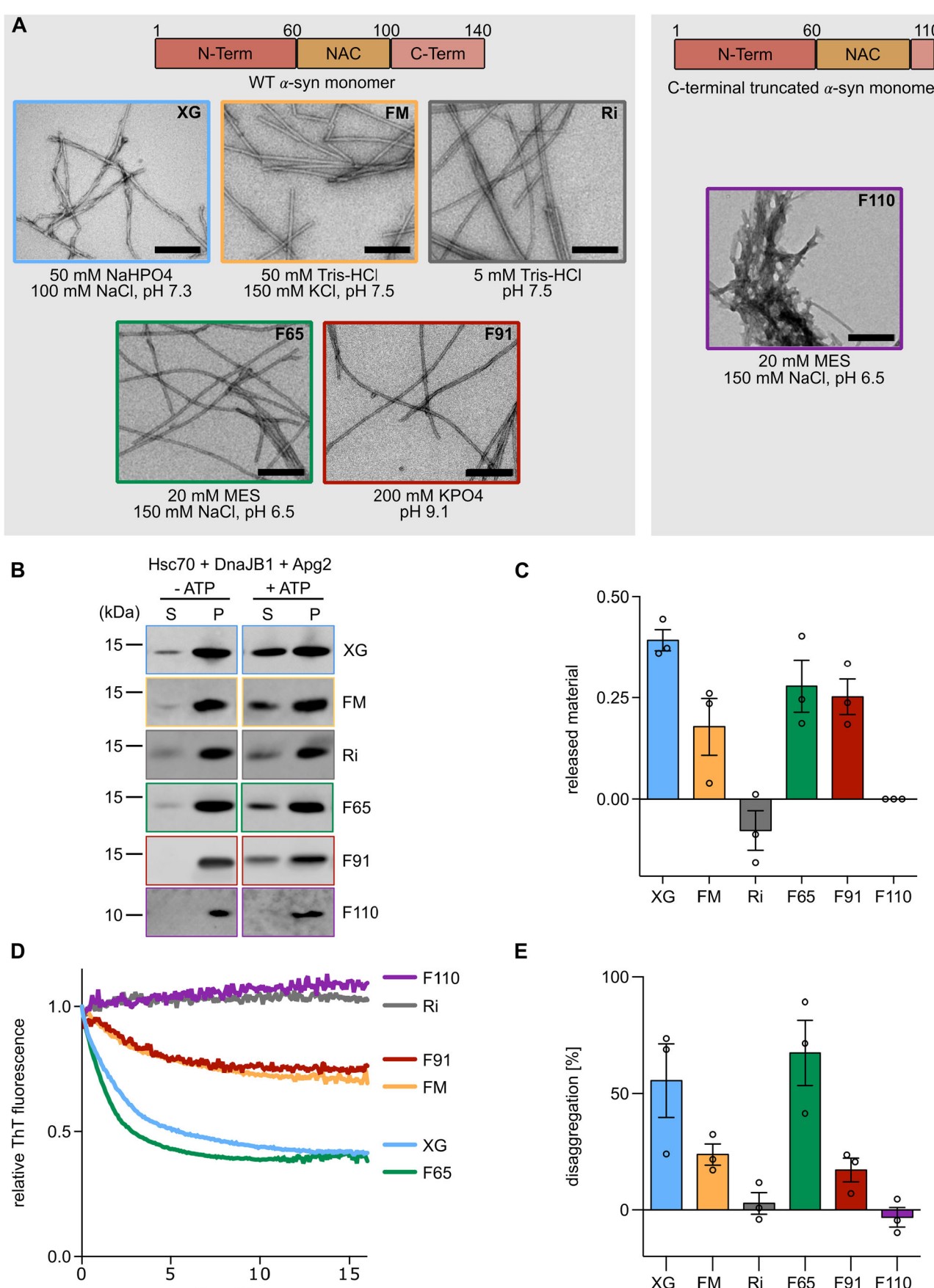

◀ **Figure 1.** **α-syn fibrillar polymorphs show differences in disaggregation kinetics by the human Hsp70 chaperone machinery.**

(A) Negative stain EM images of α-syn fibrillar polymorphs XG, FM, Ri, F65, and F91 (left) and the C-terminally truncated F110 (right) (scale bar 200 nm). The corresponding α-syn sequence is indicated above. (B) Representative immunoblots of supernatant (S) and pellet (P) fractions after a 16 h disaggregation reaction of α-syn polymorphs incubated with the active (+ATP) and inactive (−ATP) chaperone machinery (Hsc70, DnaJB1, Apg2). (C) Quantification of released α-syn protein in the supernatant as a fraction of the total (P + S) after 16 h disaggregation by the active chaperone machinery. (D) Representative data of a Thioflavin T (ThT)-based disaggregation assay of the polymorphs (XG, blue; F91, red; F65, green; FM, yellow; Ri, gray; F110, purple) by the human Hsc70 chaperone machinery. (E) Percentage of disaggregation, calculated from ThT assays, after 16 h disaggregation of the different fibrillar polymorphs by the Hsc70 machinery. Data are mean ± s.e.m. Individual datapoints represent the mean of three technical replicates for independent biological replicates (fibril batches). Source data are available online for this figure.

## Differences in polymorph structures perturb interactions with the chaperone machinery

The differences observed in chaperone-mediated disaggregation efficiencies may be explained by differences in the thermodynamic stability of the different fibril conformations, or the (in-)ability of chaperones to engage the substrate in a productive manner.

α-Syn fibrils exhibit differential stability at low temperature (Kim et al, 2008; Ikenoue et al, 2014; Bousset et al, 2013), where hydrophobic interactions are weakened, thus allowing to discriminate fibril conformations based on their thermodynamic stability. The relative stability of the polymorphs, based on their persistence at cold temperatures, serves as a broad discriminator between conformations that could or could not be disaggregated by the Hsc70 machinery (Fig. EV1A–C), with the F110 and Ri conformations both most persistent at 4 °C and resistant to chaperone-mediated disaggregation. However, resistance to cold temperatures did not readily explain the differences in disaggregation efficiency observed for the F65, XG, FM, and F91 polymorphs.

A stronger correlation was observed between the affinity of the co-chaperone DnaJB1 for the different α-syn polymorphs, determined by fluorescence anisotropy, and the disaggregation efficiency, with the Ri conformation as a notable outlier ($r = -0.97$, excluding Ri) (Fig. 2A,B). The affinity of the initial recognition of the amyloid substrate by DnaJB1 was found to differ up to sixfold between α-syn fibril conformations, ranging from 200 nM to 1.5 µM. The efficiently disaggregated XG and F65 polymorphs clustered at the lower end of this range, while DnaJB1 showed a reduced affinity for the more disaggregation-resistant substrates FM, F91, and F110.

We next asked whether reduced recognition by DnaJB1 resulted in compromised recruitment of Hsc70 to the fibrils. Dissociation constants for DnaJB1-induced binding of Hsc70 to α-syn fibrils, determined by fluorescence anisotropy, spanned one order of magnitude, ranging from 140 nM for F65 to 1.5 µM for F110 (Fig. 2C,D). The correlation between Hsc70 binding affinity and disaggregation activity was, however, less pronounced than for DnaJB1 ($r = -0.88$, excluding Ri). While the extremes in affinity corresponded to the fibril polymorphs most efficiently disaggregated (F65) and resistant to disaggregation (F110), Hsc70 affinity alone could not explain the differences in disaggregation efficiency between the FM and XG polymorphs.

To test whether the differences in disaggregation efficiency observed reflect simply a lower chaperone occupancy of some of the polymorphs (due to lower chaperone affinities) under our experimental conditions, the ThT-based disaggregation assay was repeated with twofold higher concentrations of chaperones. While increasing chaperone abundance accelerated the rates of the disaggregation reactions, it did not result in a lower endpoint ThT fluorescence for any of the α-syn fibrillar polymorphs (Fig. EV1D). Thus, the resistance to disaggregation of the F110 and Ri polymorphs, or the poor performance of the Hsc70 chaperone machinery on the FM and F91 polymorphs, cannot be surpassed simply by increased chaperone abundance. Rather, the reduced chaperone affinity for these conformations appears to reflect differences in the binding mode or positioning of the chaperones.

The three-dimensional structure of the α-syn amyloid polymorphs is thus likely to alter the accessibility of preferred chaperone binding sites in α-syn, resulting in differences in chaperone affinities and/or binding modes. The observation that DNAJB1 affinity correlates strongest with disaggregation efficiency is consistent with its critical role in the functional recruitment of Hsc70 into higher-order assemblies (Faust et al, 2020) that are key to generating the forces required for amyloid disaggregation (Wentink et al, 2020; De Los Rios et al, 2006; Goloubinoff and De Los Rios, 2007). In other instances, such as for the Ri conformation, it is likely that the inherent stability of the amyloid fibrils prevents disaggregation even when the Hsc70 machinery can successfully engage the substrate.

## Polymorph structure influences the pathway of Hsc70-mediated amyloid disaggregation

Recent studies have proposed different pathways of Hsc70-mediated disaggregation of α-syn fibrils (Schneider et al, 2021; Franco et al, 2021; Gao et al, 2015; Beton et al, 2022). Time-resolved atomic force microscopy and microfluidic diffusional sizing experiments indicated that disaggregation primarily proceeds via cooperative extraction of monomers from fibril ends, resulting in bursts of depolymerization. Instances of fibril fragmentation have, however, also been documented (Beton et al, 2022; Gao et al, 2015), although the relative frequency of these events remains uncharacterized. We hypothesize that these conflicting findings may be explained by differences in the amyloid structure of the fibrils studied. We therefore set out to characterize the reaction products formed during the disaggregation of the different α-syn amyloid conformations to document the preferred disaggregation pathway for each of the α-syn polymorphs.

Fluorescently labeled fibrils were incubated with the active chaperone machinery for 20 min, 2 h, or 16 h, and the reaction products were analyzed by sucrose density gradient centrifugation to determine the composition of products at different timepoints of the disaggregation reaction. Introduction of the fluorophore, post aggregation, did not change the morphology or efficiency of disaggregation of the fibrils (Fig. EV2A–C), and labeling efficiencies were kept purposefully low (around 5% of α-syn monomers in the fibrils) to avoid fluorescence quenching in the fibrillar

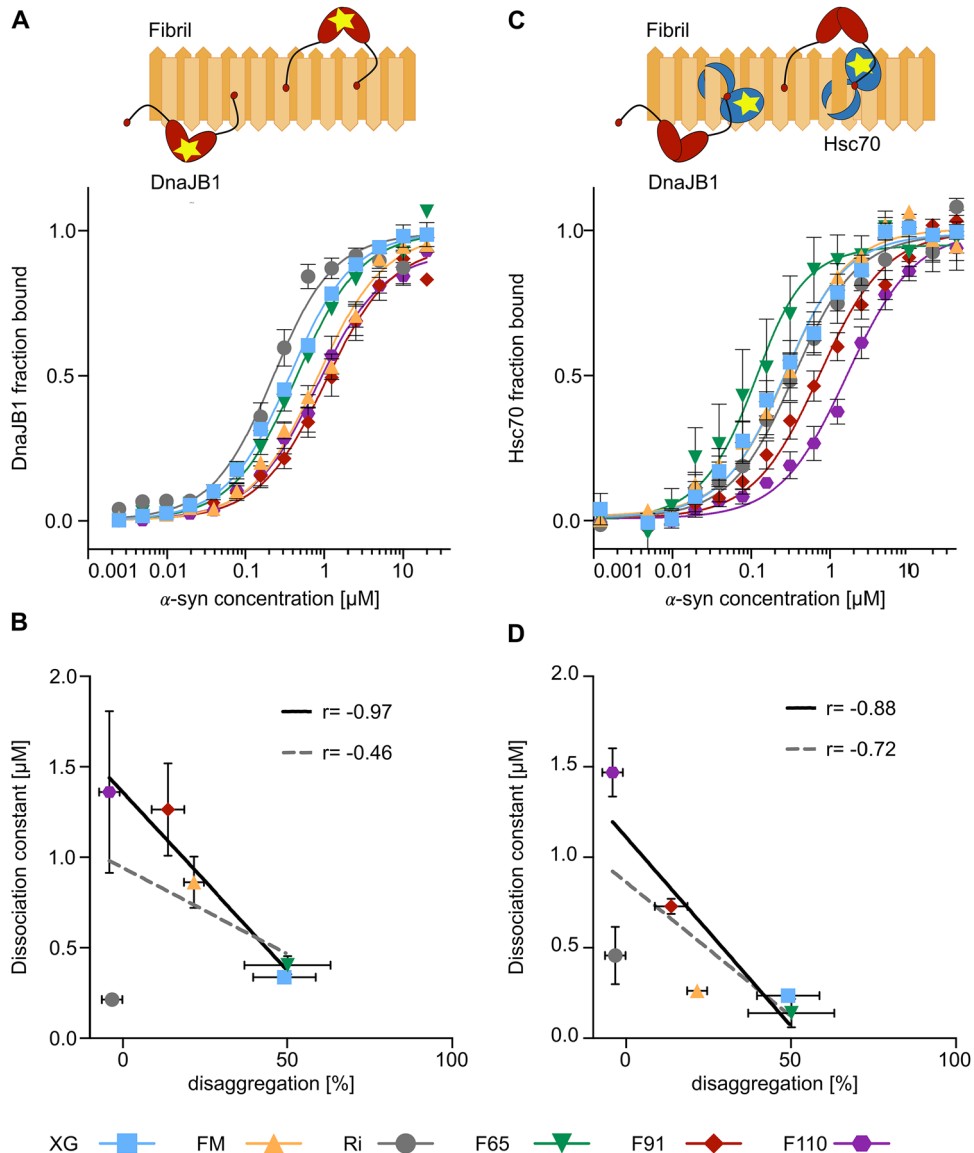

**Figure 2. Differences in polymorph structures affect interactions with the chaperone machinery.**

(A) Steady-state anisotropy titration of AF-594-labeled DnaJB1 with different concentrations of α-syn fibril polymorphs (XG; blue, FM; orange, Ri; gray, F65; green, F91; red, F110; purple). (B) Correlation of experimentally determined DnaJB1 dissociation constants and disaggregation efficiency (as determined in Fig. 1E) including (dotted line) and excluding (solid line) polymorph Ri. (C) Steady-state anisotropy titration experiment of AF-488-labeled Hsc70 in the presence of DnaJB1 and 2 mM ATP from α-syn fibril polymorphs. (D) Correlation of experimentally determined Hsc70 dissociation constants and disaggregation efficiency (as determined in Fig. 1E) including (dotted line) and excluding (solid line) polymorph Ri. Data are mean ± s.e.m. of at least three biological replicates. Source data are available online for this figure.

state (Fig. EV2D). Fluorescence intensity can thus be used as a quantitative read-out for the abundance of α-syn protein in the various fractions of the sucrose density gradient. Untreated fibrils of all polymorphs showed a sharp peak in fractions 12–20 (Fig. EV2E) corresponding to the highest sucrose concentration. Monomeric α-syn was instead found in fractions 0–2 (Fig. EV2F) at the top of the density gradient. Components of the chaperone machinery did not contribute to the fluorescence intensity signal (Fig. EV2G).

As a negative control, fibrils were incubated with the chaperone machinery in the absence of ATP for 16 h (Figs. 3A and EV2H, "No Disagg"). These non-disaggregated fibrils were found in the bottom

fractions of the density gradient corresponding to large, unprocessed fibrillar material. After 20 min incubation with the active disaggregation machinery (with ATP), the polymorph F65 showed a major shift of up to 60% of α-syn fluorescence into fractions 3–10. As fractions 0–2 were found to correspond to the α-syn monomer, fractions 3–10 contain a heterogenous range of species of intermediate oligomeric state, in molecular weight distinct from fibrils and α-syn monomer. This peak thus likely corresponds to shortened fibrils, smaller fibril fragments or oligomers produced during Hsc70-mediated disaggregation. The fluorescence signal in fractions 3–10 is reduced with increasing reaction times, with a corresponding increase in fluorescence intensity in the monomer

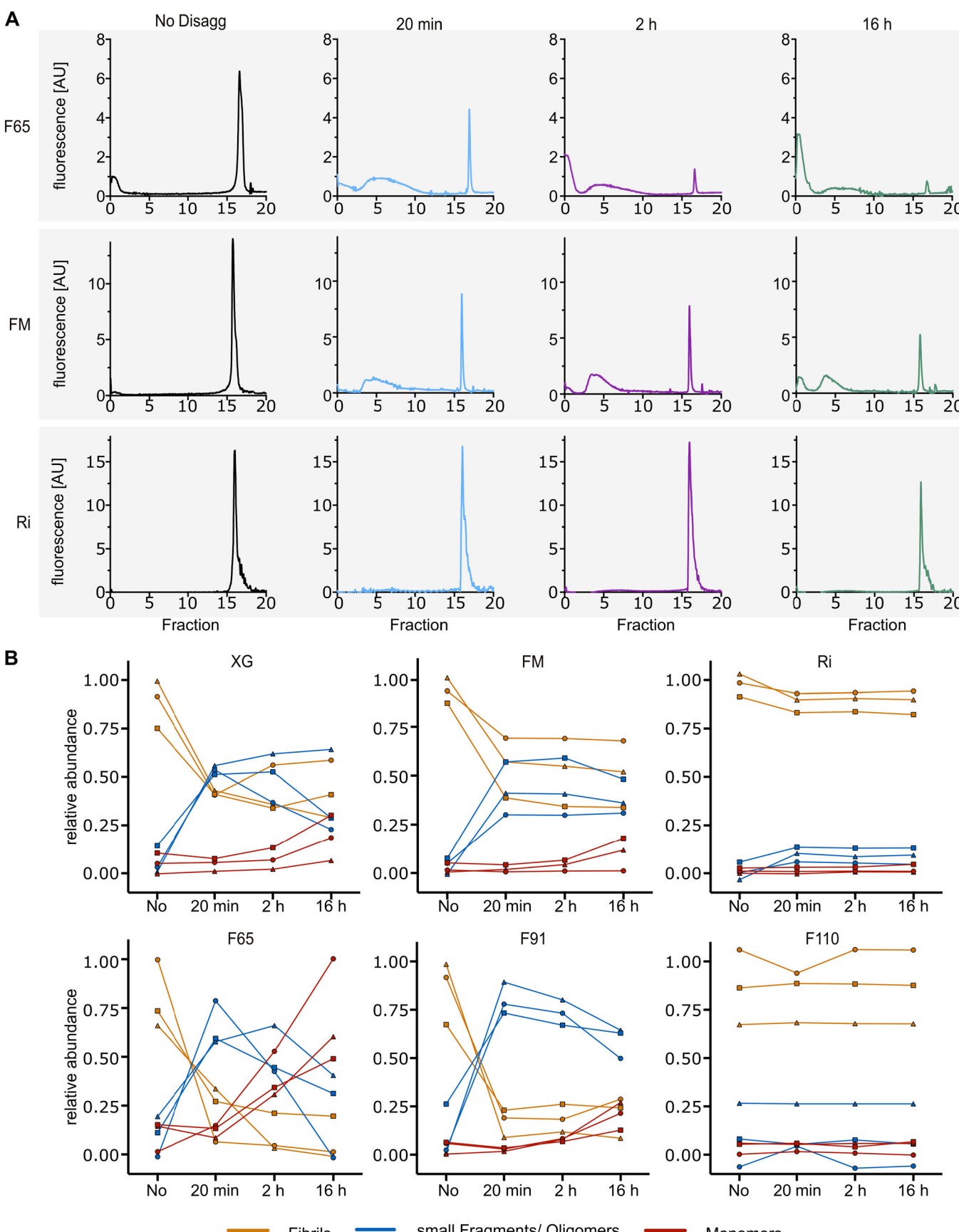

**Figure 3. Fibrillar fragments accumulate to various degrees during chaperone-mediated disaggregation of α-syn polymorphs.**

(A) Representative sucrose density gradient (10–85%) profile (AF555 fluorescence signal) of the reaction mixture after incubation of AF555-labeled fibrils with the active chaperone machinery (Hsc70, DnaJB1, Apg2, +ATP) for 20 min, 2 h, 16 h, and incubation with the inactive machinery (No Disagg, −ATP) of polymorphs F65, FM, and Ri. (B) Relative abundance of fibrils (orange), small fragments/oligomers (blue), and monomers (red) in the reaction mixtures of the specified polymorphs (XG; FM; Ri; F65; F91; F100) at the indicated timepoint during the disaggregation reaction. The relative abundance was calculated by integration of the monomer (fractions 0–2), small fragment/oligomer (fractions 3–10), or fibril peaks (fractions 12–20) of the sucrose gradient plot divided by the total. Three individual biological replicates (△, □, ○) are plotted with a connecting line. Source data are available online for this figure.

fractions 0–2 (Fig. 3A), indicating a conversion of the shortened fibrils/oligomers to monomers as the disaggregation reaction proceeds.

The polymorphs FM, XG, and F91 display similar trends, with a decrease in fibril peak intensity concomitant with a rapid conversion into shorter α-syn fibrils/oligomers and monomers over time (Figs. 3A and EV2H). In contrast, the sucrose density gradient results for Ri and the C-terminal truncated mutant F110 presented no changes in size distribution over 16 h with all fluorescence intensity found in fractions 12–20, corresponding to unprocessed fibrils (Figs. 3A and EV2H). These polymorphs are thus indeed not processed by the Hsc70 chaperone machinery.

The sucrose density gradient results for the FM and F91 fibrils diverge from the ThT assays, which reported only limited disaggregation activity. Quantification of the relative abundance of the monomer (fractions 0–2), oligomers and shortened fibril (3–10) and fibril (12–20) fractions for the FM and F91 polymorphs at the different timepoints (Fig. 3B) indicate that, while fluorescence intensity in the fibrillar fractions rapidly decreased and stabilized from 20 min onwards around 55% and 20% of the starting fluorescence intensities, respectively, α-syn monomers accumulate very slowly and are absent (<5%) at 20 min. The main products of chaperone-mediated disaggregation of the FM and F91 polymorphs, found in fractions 3–10, are thus likely to be fibril fragments, produced as a result of fibril fragmentation. Such fragments would retain ThT binding capacity and sediment upon centrifugation, explaining the apparent poor disaggregation efficiencies detected by previous experiments. The Hsc70 machinery can thus in fact process the FM and F91 α-syn fibrillar polymorphs efficiently but appears to rely on a disaggregation pathway that includes fibril fragmentation.

In contrast, the distribution of reaction products of the F65 polymorph are broadly consistent with a disaggregation mechanism driven primarily by depolymerization of monomers from fibril ends (Franco et al, 2021; Schneider et al, 2021; Gao et al, 2015; Beton et al, 2022), with a gradual reduction in the fibrillar fraction over time and the appearance of monomers at early timepoints (Fig. 3B). As a consequence, the oligomer/fragment species are less abundant and more broadly distributed, with the average size shrinking as a function of time during the disaggregation of F65 as reflected by their position in the gradient (Fig. 3A). The XG fibrils display an intermediate behavior, with rapid processing of the initial fibril material into smaller oligomeric species followed by relatively efficient conversion to monomers (Fig. 3B).

We conclude from our results that fibril fragments and/or oligomers accumulate to different extents during the chaperone-mediated disaggregation of distinct α-syn fibrillar polymorphs. The relative abundance of these species is likely to reflect the different sensitivities of the polymorphs to depolymerization and/or fragmentation.

## Disaggregation of α-syn fibrils produces seeding competent species in vitro

Pathological protein aggregates associated with neurodegenerative disease have been hypothesized to propagate in a prion-like fashion, where initial amyloid seeds can serve as conformational template to promote further aggregation (Tittelmeier et al, 2020; Marrero-Winkens et al, 2020; Braak et al, 2003; Jucker and Walker, 2018). The differential accumulation of oligomer/fragment species during chaperone-mediated disaggregation among α-syn polymorphs thus may alter their ability to propagate and spread from cell to cell in a disease context.

To evaluate the potential role of chaperone-mediated disaggregation in the prion-like amplification of α-syn seeds, the seeding capacity of the disaggregation reaction products was characterized for the F65 and FM polymorphs as representative examples of amyloid conformations where disaggregation occurs primarily by depolymerization or where fragmentation is prevalent, respectively. The seeded aggregation of α-syn monomers was monitored in vitro by ThT fluorescence (Fig. 4A,B). The untreated fibrils seeded the aggregation of α-syn monomers efficiently. Incubation with an inactive chaperone machinery ("No disagg") resulted in a factor 2 decrease in seeding rates, consistent with an expected prevention of aggregation activity of the Hsc70 machinery (De Mattos et al, 2020). The prior disaggregation of FM and F65 fibrils for 16 h did not reduce the seeding capacity further (Fig. 4A,B). This is surprising as the ATP-driven disaggregation activity of Hsc70 processes 40% and 75% of the starting material, respectively, over 16 h (Fig. 3B) and a corresponding drop in the seeding capacity might be expected if the residual unprocessed fibrils were the only source of seeds in these reaction mixtures. Disaggregation of both the FM and F65 polymorphs thus produces a novel seeding competent species.

To identify the novel seeding competent species in the reaction mixtures, the total disaggregation reactions were fractionated by centrifugation (Fig. 4C). Residual unprocessed fibrillar material was depleted from the disaggregation products by low-speed centrifugation (15 min at $3600 \times g$). Higher centrifugation speeds (30 min at $435{,}630 \times g$) additionally separated monomers from smaller fragments/oligomers as confirmed by electron microscopy (EM) and sucrose density gradients (Fig. 4D and Appendix Fig. S2A–C). For both polymorphs, the supernatant of the low-speed centrifugation step retained a large fraction of the seeding capacity of the total disaggregation mixture. The high-speed centrifugation sample was, however, seeding incompetent (Fig. 4E,F). In contrast, the fractionation of fibrils treated with the inactive chaperone machinery showed a total depletion of seeding capacity even after the first low-speed centrifugation step (Fig. 4G,H). The new seeding competent species created through the disaggregation of α-syn

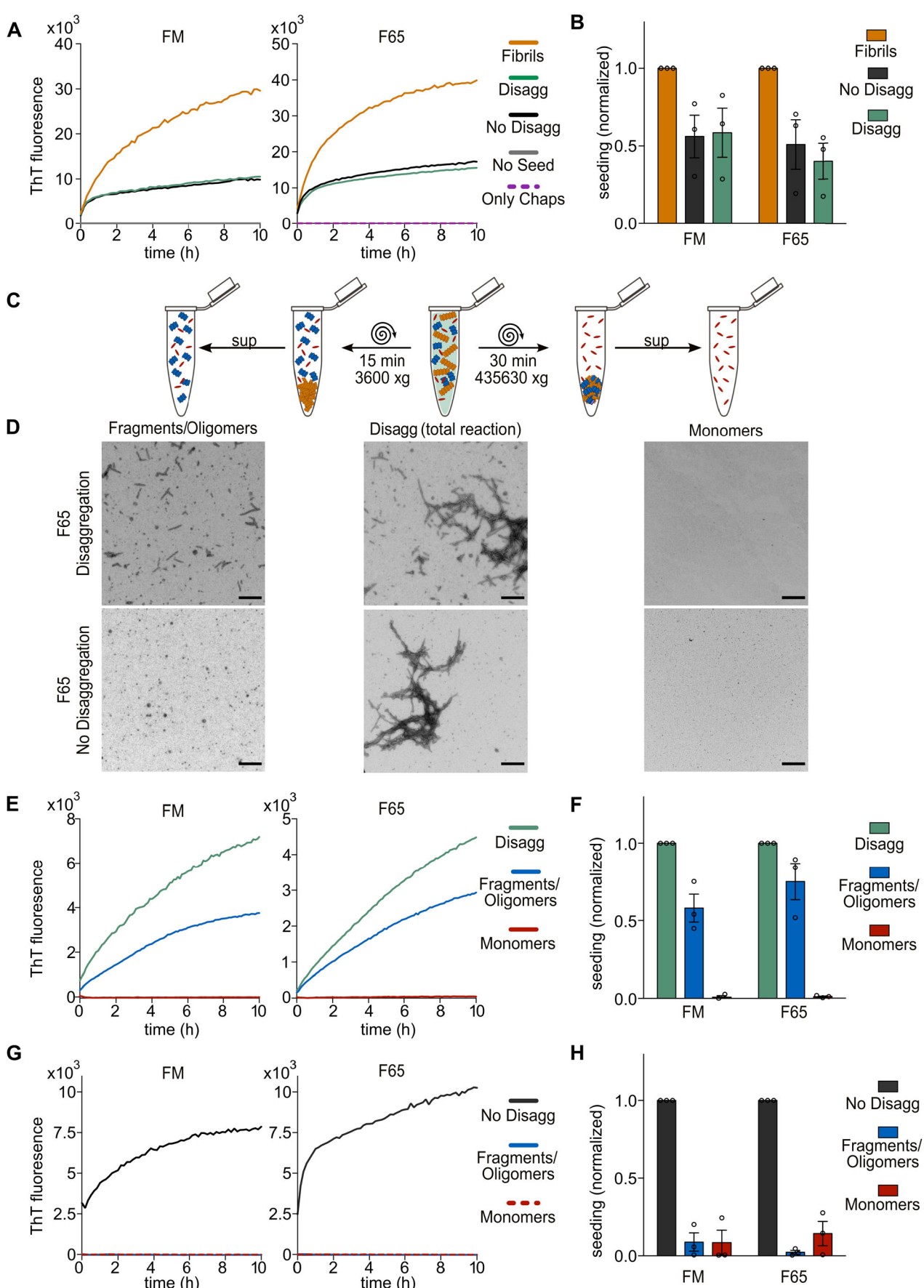

**Figure 4.  Disaggregation of fibrils produces seeding competent fragments in vitro.**

(A) Representative aggregation curves of α-syn monomers seeded with Fibrils (orange); No Disaggregation (black), Disaggregation (green), No seed (gray), and only chaperones (purple) samples for the polymorphs FM (left) and F65 (right). (B) Quantification of the initial rate of the seeded aggregation reactions in (A), normalized to the initial rate of the fibril only reaction. (C) Centrifugation procedure to separate different disaggregation reaction products. A total disaggregation reaction is centrifuged at $3600 \times g$ for 15 min, fibrils (orange) are separated in the pellet and small fragments/oligomers remain in the supernatant (left). By centrifugation of the total reaction at a higher speed ($435,630 \times g$) for 30 min, small fragments/oligomers (blue) and fibrils are pelleted and only monomers (red) remain in the supernatant (right). (D) Representative EM images of a disaggregation and No disaggregation reaction (total reaction; middle), small fragments/oligomer fraction (left) and monomer fraction (right); scale bar 500 nm. (E) Representative aggregation curves of α-syn monomers seeded with a total disaggregation reaction (green) and fractionated small fragments/oligomer (blue) and monomer (red) fraction of polymorph FM (left) and F65 (right). (F) Quantification of the initial rate of the seeded aggregation reactions in (E), normalized to the initial rate of the total disaggregation reaction. (G) Representative aggregation curves of α-syn monomers seeded with fractions of a non-disaggregated (−ATP) reaction mixture (total, green; fragments/oligomer, blue; monomer, red). (H) Quantification of initial rate of seeded aggregation reactions as shown in (G) normalized to the total disaggregation reactions. Data are mean ± s.e.m. Individual datapoints represent the mean of two technical replicates for three independent biological replicates. Source data are available online for this figure.

fibrils thus likely comprise smaller fibril fragments or oligomers that retain features of the amyloid conformation.

The initial aggregation rate of α-syn monomers seeded with the F65 low-speed centrifugation fraction is reduced by only 25% compared to the total reaction mixture (Fig. 4E,F), consistent with 75% of the starting material having been processed and now found in this soluble fraction (Fig. 3B). This fraction contains a mixture of released monomers and fibril fragments. Disaggregation by depolymerization would result in a lower total mass of fibrils over time, but broadly preserves the total number of fibril ends in the sample that can serve as a template for amyloid fibril growth until fibrils have been disaggregated to completion. The observed 75% seeding capacity is thus at the upper limit of what one might expect for fibril disaggregation based solely on depolymerization, but would imply that very few fibrils have been disaggregated to completion, or that at least some fragmentation occurs.

In contrast, the same fraction of fragments/oligomers of the FM polymorph, containing only 40% of the starting fibrillar material (Fig. 3B), accounted for 60% of the seeding capacity of the total reaction mixture (Fig. 4E,F). This suggests that the newly formed products are per monomer equivalent more seeding competent than the original starting material. This is consistent with our observation that FM fibrils are sensitive to fragmentation during disaggregation, which would result in a growing number of smaller fibrils, each capable of templating further protein aggregation. This, despite an overall decrease in total aggregate mass by Hsc70-mediated disaggregation.

Despite these differences, both α-syn polymorph samples retained a large fraction of their in vitro seeding capacity after 16 h of disaggregation, suggesting that unless fibrils are fully dissolved to monomers, Hsc70-mediated disaggregation may have limited protective function against further protein aggregation. Nevertheless, the anti-aggregation activity of the Hsc70 machinery observed in the reactions without ATP ("No disagg") compared to the untreated fibrils ("Fibrils") in Fig. 4A,B dominates these in vitro seeding reactions, indicating a net protective effect of the presence of the Hsc70 machinery on both α-syn polymorphs in vitro.

## Disaggregation reaction products trigger intracellular aggregation of α-syn in human cells

We next assessed the seeding capacity of the Hsc70 machinery-derived α-syn species from the different polymorphs in a cellular context. To this end, we used a well-established HEK293 biosensor cell line (Sanders et al, 2014; Tittelmeier et al, 2022) that stably expresses aggregation-prone A53T mutant α-syn, C-terminally fused to yellow fluorescent protein (α-syn A53T-YFP) (Fig. 5A). Before exposure to recombinant α-syn fibrils, the α-syn A53T-YFP reporter exhibited diffuse cytosolic fluorescence (Fig. 5B, panel "untreated"). Once exposed to the α-syn polymorphs, we observed the accumulation of fluorescent α-syn A53T-YFP foci within the cells, reflecting intracellular aggregation of the reporter seeded by inoculated fibrils (Fig. 5A,B). To reduce potential differences in cell surface binding and uptake efficiency between structurally distinct α-syn fibrillar polymorphs, fibrils were introduced into the cytosol with Lipofectamine. Except for the Ri conformation, all α-syn fibrillar polymorphs induced intracellular aggregation of α-syn in up to 15% of cells (Figs. 5A,B and EV3A,B).

The seeding capacity of the FM, F65, F91, and XG polymorphs is reduced by 2–4-fold after incubation with the chaperone machinery in the absence of ATP ("No Disaggregation", Figs. 5B,C and EV3A,B). In contrast, the seeding capacity of the F110 polymorph was unaltered by pre-incubation with the Hsc70 chaperone machinery ("No Disaggregation", Fig. EV3A,B). We hypothesized that the observed prevention of aggregation activity could be attributed to a direct interaction of chaperones with the fibrils. Incubation with individual components of the Hsc70 chaperone machinery identified DNAJB1 as the key chaperone responsible for reducing the seeding capacity of XG α-syn fibrils to 30% of the untreated fibrils (Fig. EV4). In contrast, no combination of chaperones affected the seeding capacity of the F110 polymorph, consistent with the reduced affinity of the Hsc70 chaperone machinery for this polymorph (Figs. EV4A and 2).

Incubation of α-syn fibrillar polymorphs sensitive to disaggregation with the active chaperone machinery increased their overall seeding capacity by 3–4-fold compared to their seeding propensity upon incubation with the chaperones in the absence of ATP ("Disaggregation" versus "No Disaggregation", Fig. 5B,C). Thus, despite an overall decrease in amyloid content ranging from 40% to 80% after disaggregation in all of these reactions (Fig. 3B), the resulting material approximates the seeding capacity of the fibrils prior to exposure to chaperones. This seeding ability could again be attributed primarily to small oligomers/fragments produced during disaggregation, rather than the residual unprocessed fibrils, based on fractionation of monomer, oligomer and fibril fractions by centrifugation (Fig. 4C). The cellular context thus negates in large part the protective effects of the chaperone machinery (as observed in the "No disaggregation" samples, Fig. 5) while amplifying the

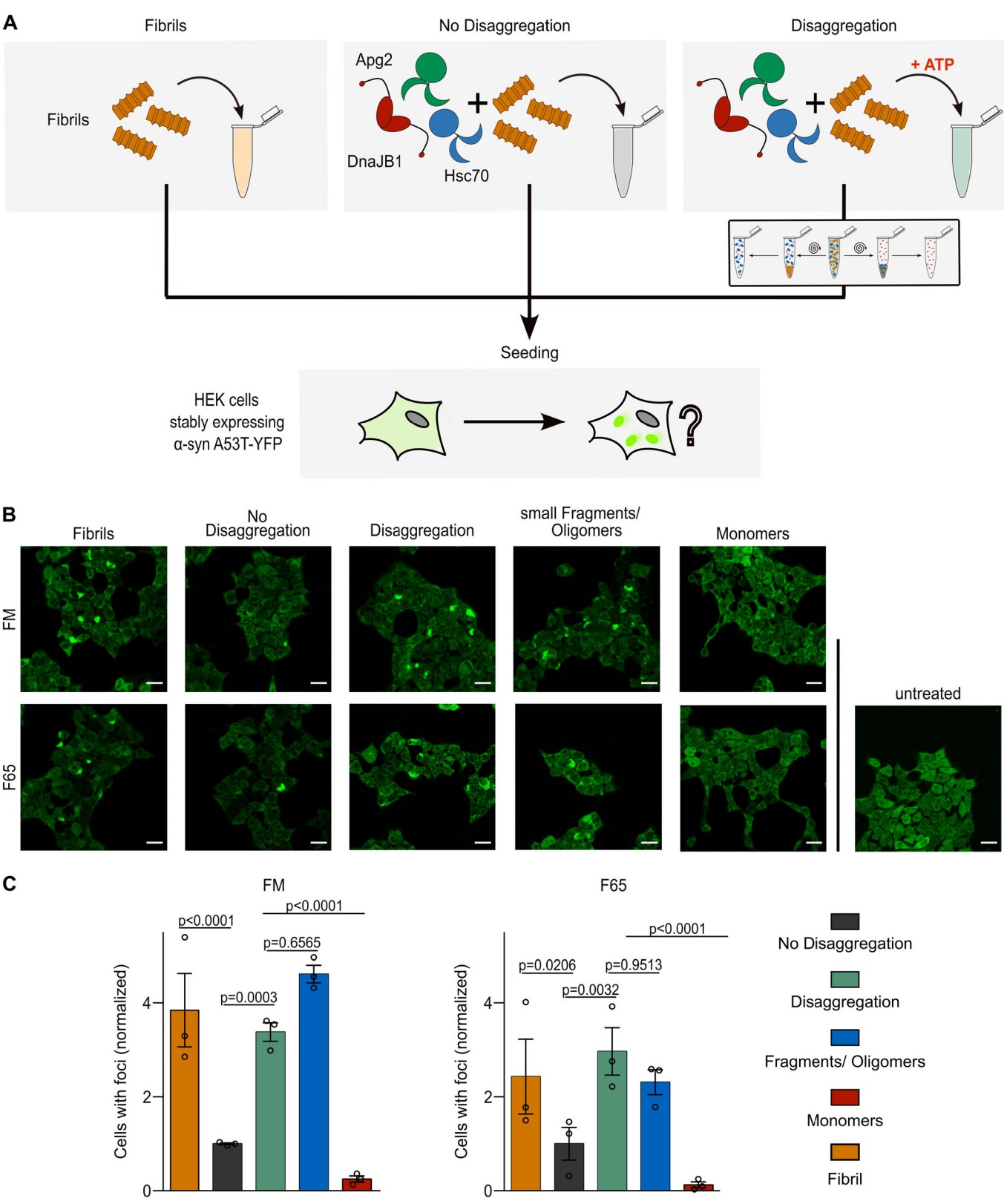

**Figure 5.  Disaggregation reaction products trigger foci formation in a human cell model.**

(A) Experimental setup of an in cellulo seeding assay. Three different reaction mixtures are analyzed after 16 h incubation at 30 °C: fibrils only (Fibrils), fibrils incubated with the chaperone machinery (Hsc70, DnaJB1, Apg2) in the absence (No Disaggregation), and presence of ATP (Disaggregation). In addition, the disaggregation sample is further fractionated by differential centrifugation as described in Fig. 4C to generate "Fragments/Oligomers" and "Monomers" fractions. The reaction mixtures are added as seeds to HEK293T cells stably expressing α-syn A53T-YFP, and the number of cells with fluorescent foci is counted. (B) Representative fluorescence microscopy images of reporter HEK293T cells seeded with treated fibrils of polymorphs FM and F65. Untreated cells are shown in the lower right (scale bar 20 μm). Fibrils only, fibrils incubated with chaperones in the absence (No Disaggregation) and presence of ATP (Disaggregation), as well as small fragments/oligomer and monomer fractions of the disaggregation reaction separated by centrifugation as described in Fig. 4. (C) Quantification of the percentage of cells with foci normalized to the No disaggregation sample (No Disaggregation, black; Disaggregation, green; small fragments/oligomers, blue; monomers, red; fibrils, orange). Data are mean ± s.e.m. of three biological replicates. Statistical analysis was performed by nonparametric two-way ANOVA with pairwise comparisons of estimated marginal means with Tukey correction for multiple comparisons. Exact p values are indicated in (C). Source data are available online for this figure.

potential threat associated with the production of fibrillar fragments/oligomers during chaperone-mediated disaggregation as compared to in vitro experiments.

## Discussion

In this study, we show that conformationally distinct fibrillar polymorphs prepared from recombinant human α-syn protein exhibit differential susceptibility to disaggregation by the human Hsc70 chaperone machinery. We observe that high thermodynamic stability of the specific conformation, as deduced from a lack of depolymerization at low temperatures, is associated with a complete resistance to chaperone-mediated disaggregation under the tested conditions suggests there is a thermodynamic limit to the ability of the chaperone machinery to disaggregate amyloid fibrils. Furthermore, chaperone affinity, in particular DNAJB1, correlates well with disaggregation efficiency but furthermore appears to determine the type of species that accumulate during disaggregation, with high-affinity substrates such as the F65 fibrils seemingly disaggregated primarily via depolymerization into monomeric α-syn, while more fragmentation occurred with less ideal target conformations, such as the FM and F91 polymorphs (Fig. 6A).

The observed differences in chaperone affinity likely reflect the accessibility and geometry of important chaperone binding sites in the N- and C-terminus of α-syn (Fig. 6B) (Redeker et al, 2012; Wentink et al, 2020; Burmann et al, 2020; Nury et al, 2015). DnaJB1 is seemingly most sensitive to these differences, consistent with the report that its affinity for the aggregated state of α-syn is driven by avidity effects (Wentink et al, 2020), which would be affected by subtle differences in distance or orientation of proximal binding sites. Little structural information is available about this region as it remains highly unstructured in all fibrillar states. The differences in DnaJB1 affinity for the α-syn polymorphs are, however, consistent with those of a C-terminal-specific antibody 10D2 (Landureau et al, 2021), which binds the Ri conformation more strongly than the FM and F91 fibrils, suggesting reduced accessibility or potential for multivalent binding in these conformations.

Similarly, the subsequent recruitment of Hsc70 is likely to be affected by (1) the positioning of DnaJB1 and (2) the degree of structure of the α-syn N-terminus where important Hsc70 binding sites reside. Key variations in the degree of structure in this region were described for the different studied α-syn polymorphs (Fig. 6B) (Guerrero-Ferreira et al, 2019; Landureau et al, 2021; Verasdonck et al, 2016). For example, in the Ri polymorph the N-terminus is highly ordered, potentially altering the binding mode of Hsc70 and

thus its ability to assemble into disaggregation competent complexes.

In cryo-EM structures of the FM polymorph (Fig. EV5A), residues 16–24 are found associated with the fibrillar core (Guerrero-Ferreira et al, 2019), rendering the N-terminus less accessible to chaperone binding. The N-terminus of the other, more disaggregation-resistant polymorph F91 shows a high degree of protection against hydrogen–deuterium (HD) exchange (Landureau et al, 2021), despite lacking obvious secondary structure (Verasdonck et al, 2016). In contrast, the XG polymorph displays a higher degree of structural variation in the N-terminus, as visible in recent cryo-EM structures that contain both partially structured and unstructured N-termini (Monistrol et al, 2025) and reflected in HD exchange experiments where the N-terminus generally exchanges rapidly, with some locally protected sites (Fig. EV5A,B). Based on the available structural data of the polymorphs, a trend therefore emerges where limited solvent accessibility of the N-terminus such as seen in the Ri fibrils, and to a lesser extent in the F91 polymorph, reduces susceptibility to disaggregation.

These findings echo previous work on the yeast Sup35 prion, where structural polymorphs display different susceptibilities to fragmentation by the Hsp70-Hsp100 bi-chaperone system (Ohhashi et al, 2018), which correlated with the exposure of key chaperone binding sites (Shen et al, 2024). It should be noted that within this study we have only explored the sensitivities of the different α-syn polymorphs to disaggregation by the specific combination of the chaperones Hsc70, DNAJB1, and Apg2, at stoichiometries that have been found to be ideal for the disaggregation of the XG polymorph (Gao et al, 2015; Wentink et al, 2020). It is therefore possible that other combinations of co-chaperones would be more suited to tackle alternative conformations of α-syn fibrils.

Different sensitivities of α-syn polymorphs to disaggregation have potentially important implications for disease progression. Disaggregation pathways where depolymerization is inefficient would lead to the accumulation of fibril fragments, which we show are coupled to a higher in vitro seeding capacity. Fibril conformations that predominate in disease may therefore be those that are most sensitive to fragmentation, boosting their amplification and propagation. Furthermore, brain cells where chaperone-mediated disaggregation is most active may be most susceptible to pathology onset and progression when compared to neighboring cells that survive.

C-terminal proteolysis of α-syn is frequently observed in Lewy bodies (Zhang et al, 2017; Li et al, 2005; Liu et al, 2005). We could demonstrate that a missing C-terminus leads to resistance to

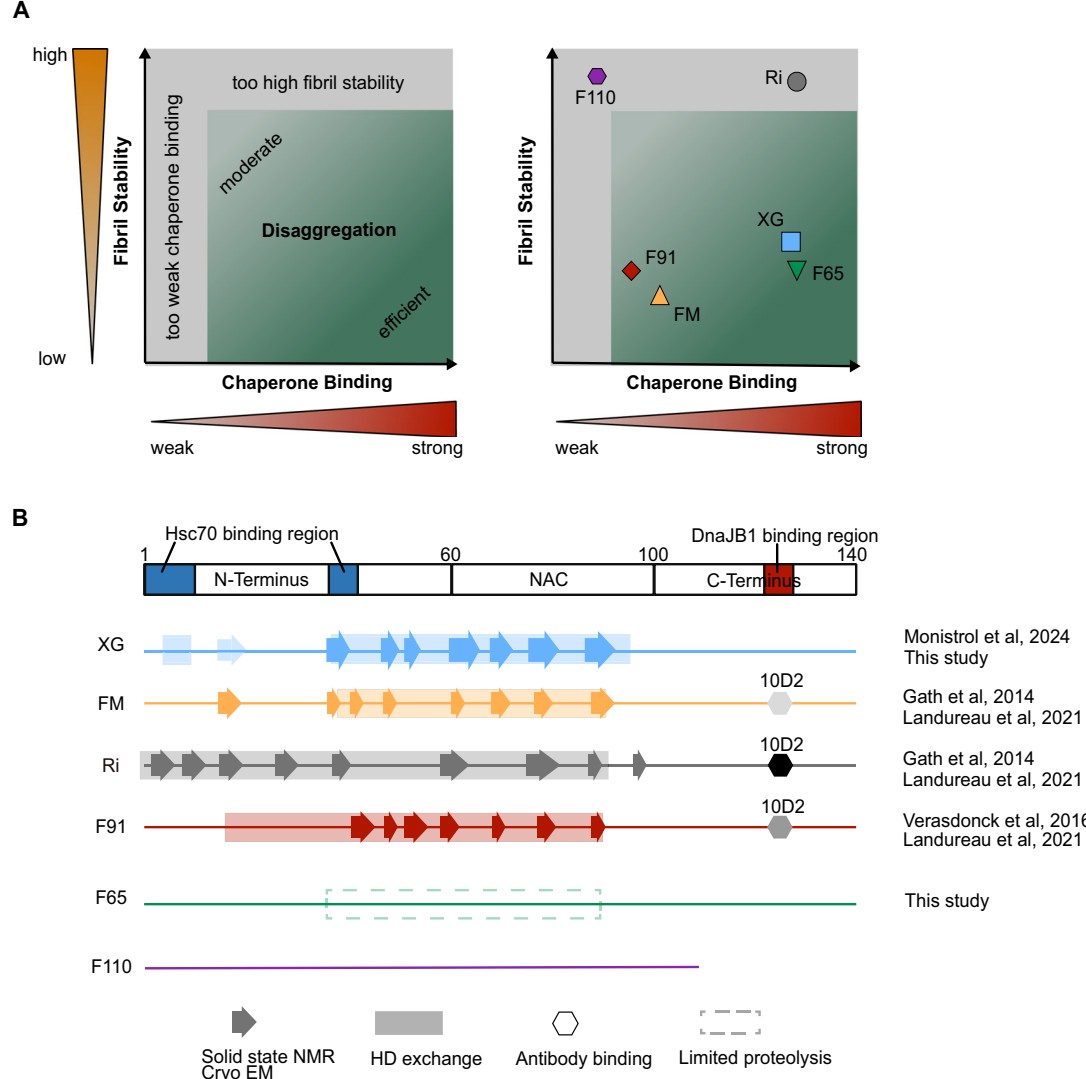

**Figure 6. Disaggregation efficiency depends on a combination of chaperone binding and fibril stability.**

(A) Model of disaggregation efficiency (green) as a function of fibril stability (y-axis) and chaperone binding (x-axis). The interplay between chaperone binding to the fibrils and fibril stability has to be in the indicated green area for efficient disaggregation, whereas fibrils located in gray areas (high stability and/or low chaperone binding) are resistant to disaggregation. The distribution of fibril polymorphs analyzed in this study is shown in the plot on the right. (B) Cartoon representation of in vitro aggregated α-syn polymorph XG, FM, Ri, F91, F65, and F110. The monomeric α-syn sequence consisting of N-terminal domain, NAC-region, and C-terminal domain with proposed chaperone binding sites indicated (Hsc70, blue; DnaJB1, red) based on Wentink et al (2020) (top). Known regions of the monomeric α-syn sequence in the polymorphs that are incorporated into β-sheet structures (small arrow) or are protected from hydrogen–deuterium exchange (shaded area) are indicated. In addition, the respective binding strength of the C-terminal binding antibody 10D2 is indicated (scale from light to dark with increasing affinity). Structural information for F65 can thus far only be derived from similarity in the limited proteolysis pattern to the FM polymorph (Fig. EV1A), suggesting similar regions are buried in the fibril core. In particular, the N-terminal region varies in the degree of structure observed between the various polymorphs, coinciding with the proposed binding sites of Hsc70.

disaggregation, most likely because important chaperone binding sites are located in the flexible C-terminus of the α-syn fibrils (Wentink et al, 2020; Redeker et al, 2012). We also observed that F110 fibrils tended to bundle together as seen under EM (Fig. 1A), which would additionally hinder functional chaperone engagement. This posttranslational modification might thus transfer the protein aggregate into a state that is resistant to clearance by chaperones and result in its accumulation in patient brains. In the context of our observation that disaggregation significantly increased the intracellular seeding capacity of α-syn fibrils, this induced

resistance to disaggregation may in fact inhibit further aggregate propagation. This posttranslational modification could therefore be a protective mechanism against prion-like propagation of the disease phenotype.

Our study also highlights the emergent properties of the complex cellular environment, with the seeding capacity of disaggregation products generated in vitro (Fig. 5) strongly amplified in cell-based assays compared to in vitro experiments (Fig. 4). In this reporter cell line, the high expression levels of aggregation-prone α-syn A53T readily promote templated

aggregation based on internalized seeds, outcompeting any intracellular disaggregation processes. As a counterpoint, the effective coupling of disaggregation pathways to refolding or degradation pathways within the cellular context may result in synergies in the intracellular disaggregation of α-syn fibrils that were not explored in this study. For instance, in the case of tau fibrils, the Hsc70 machinery can act downstream of the VCP/p97 chaperone machinery. The AAA+ ATPase VCP/p97 pre-processes ubiquitinated tau fibrils by fragmentation but relies on Hsc70 action to fully dissolve existing tau aggregates (Saha et al, 2023). Such synergies may alleviate some of the potential dangers associated with fragment production during disaggregation (Nachman et al, 2020) at physiological expression levels of α-syn. It will thus be important to further investigate the balance between protective and harmful chaperone activities within the cellular context.

Taken together, the cellular chaperone machinery may differently affect the formation and clearance of α-syn amyloid fibrils in different synucleinopathies, depending on the predominant α-syn polymorph. This model makes it challenging to design a universal approach to exploiting chaperones for therapeutic applications. Nevertheless, these findings highlight the importance of considering conformational diversity when investigating the molecular basis of pathological protein aggregation and possible future therapeutics that intervene in the process.

# Methods

### Reagents and tools table

| Reagent/resource | Reference or source | Identifier or catalog number |
|---|---|---|
| **Experimental models** | | |
| *E. coli* XL1 blue - recA1 end A1 gyrA96 thi-1 hsdR1 supE44 rel A1 lac [F' proAB laclq DM15 Tn10 (Tctr)] | Aligent Technologies | 200249 |
| *E. coli* BL21 Rosetta - F- ompT lon hsdSB(rB- mB-) gal dcm pRARE (CamR) l_(DE3) | Sigma Aldrich | 70954 |
| *E. coli* BL21 Rosetta gold - F– ompT hsdS(rB – mB –) dcm+ Tetr gal endA Hte | Aligent Technologies | 230132 |
| HEK293 biosensor cell line | Yamasaki et al (2019) | |
| **Recombinant DNA** | | |
| pCA528-DNAJB1 | Rampelt et al (2012) | |
| pCA528-2G-Apg2 | Rampelt et al (2012) | |
| pCA528-Hsc70 | Rampelt et al (2012) | |
| pCA528-DNAJB1-G194C | Gao et al (2015) | |
| pCA528-Hsc70-C11-C267A-C574A-C603A-T111C | Gao et al (2015) | |
| pT7-7 α-syn -WT | Wentink et al (2020) | |
| **Antibodies** | | |
| α-syn (SNCA) monoclonal mouse IgG | Santa Cruz Biotechnologie | Sc-12767 |

| Reagent/resource | Reference or source | Identifier or catalog number |
|---|---|---|
| α-syn (SNCA) (61-95) monoclonal mouse IgG | Origene | SM6028 |
| Alkaline phosphatase coupled anti-mouse-IgG (H + L) (horse) | Vector Laboratories, Inc. | AP-2000 |
| **Oligonucleotides and other sequence-based reagents** | | |
| **Chemicals, enzymes, and other reagents** | | |
| Aprotinin | AppliChem | A2132 |
| DNAse I | Roche | 04716728001 |
| Leupeptin | AppliChem | A2183 |
| Pepstatin A | AppliChem | A2205 |
| Pyruvate kinase from rabbit | Sigma | 10128155001 |
| Phosphatase, alkaline (AP) | Roche | 10108138001 |
| Adenosine triphosphate ATP | Sigma-Aldrich | A2383 |
| Alexa Fluor 488, 594, 532 C5 maleimide | Thermofisher Scientific | A10254, A10256, A10255 |
| Alexa Fluor 555 NHS ester (succinimidyl ester) | Thermofisher Scientific | A20009 |
| Ammonium sulfate | Roth | 9218.1 |
| Bond-Breaker TCEP Solution | Thermofisher Scientific | 77720 |
| Coomassie quick stain | Protein Arc | GEN-QC-STAIN-1L |
| Disodium salt 2-hydrate, EDTA | PanReac AppliChem | 131669 |
| DL-Dithiothreitol (DTT) | Sigma | D0632 |
| DMEM | Gibco | 31966-021 |
| Gibco Fetal Bovine Serum (FBS) | Thermofisher Scientific | A5670701 |
| Gibco GlutaMAX Supplement | Thermofisher Scientific | 35050038 |
| Glycerol | Roth | 3783.5 |
| Glycine | Sigma-Aldrich | G7126 |
| Guanidine hydrochloride | Roth | 0037.1 |
| HEPES | Roth | 6763.1 |
| Imidazole | Roth | 3899.3 |
| Isopropyl-β-D-thiogalactopyranosid (IPTG) | Roth | 2316 |
| Lipofectamine2000 | Invitrogen | 11668-019 |
| Formaldehyde solution | Sigma | 1040031000 |
| Magnesium chloride hexahydrate | Roth | 3532.1 |
| Natrium chloride, NaCl | Roth | 3957.1 |
| Gibco Opti-MEM Reduced Serum Medium | Thermofisher Scientific | 31985062 |
| Gibco Penicillin-Streptomycin | Thermofisher Scientific | 15140122 |
| Phospho(enol)pyruvic (PEP) acid trisodium salt hydrate | Sigma | P7002 |
| Poly-L-Lysine | Sigma | P4707 |
| Potassium chloride, KCl | Roth | 6781.1 |
| Gibco Pyruvate | Thermofisher Scientific | 11360070 |
| SDS pellets | Roth | CN30.1 |

| Reagent/resource | Reference or source | Identifier or catalog number |
|---|---|---|
| β-Mercaptoethanol, BME | Roth | 4227.1 |
| Sucrose | Sigma-Aldrich | S0389 |
| TRIS | Roth | 4855.2 |
| Tween20 | Sigma-Aldrich | P9416 |
| **Software** | | |
| ImageJ | Wayne Rasband, National Institutes of health, USA | |
| Inkscape | InkscapeTM | |
| Office 365 | Microsoft | |
| Prism 6 | GraphPad software, Inc. | |
| **Other** | | |
| GenElute HP Plasmid Miniprep Kit | Sigma-Aldrich | NA0150 |
| TRIAXTM flow cell, mCherry, EGFP, 280 nm | Biocomp | BFC-3 |
| Piston Gradient FractionatorTM | Biocomp | |
| Gradient MasterTM | Biocomp | |
| FLUOstar, SPECTROstar OMEGA plate reader | BMG Labtech | |
| Leica TCS SP8 STED 3X | Leica | |

## Protein expression and purification

Chaperone proteins, Hsc70, DnaJB1, Apg2, and their respective mutants (DnaJB1-G194C, Hsc70-C267A-C574A-C603A-T111C) were purified as previously described by Nillegoda et al (2015). Briefly, proteins were expressed as $His_6$-Smt3 (H6-sumo) fusion proteins in *E. coli* BL21 Rosetta cells, followed by nickel affinity purification (Ni-IDA, Macherey-Nagel) upon lysis. The $His_6$-Smt3 tag was cleaved by ULP1 protease during overnight dialysis, and a second, reverse nickel affinity purification was carried out to separate the final protein from the tag, protease, and uncleaved protein.

WT α-syn was recombinantly expressed from a pT7-7 or pET14b expression vector as untagged protein in *E. coli* (DE3) Gold and purified as described before (Hoyer et al, 2002; Ghee et al, 2005).

## Fibril preparation

Purified α-syn monomers were aggregated in the corresponding buffer under continuous shaking for 7 days at 37 °C (Buffer Compositions: XG—50 mM $NaHPO_4$, 100 mM NaCl, pH 7.3, 1000 rpm; FM—50 mM Tris-HCl 150 mM, KCl, pH 7.5, 600 rpm; Ri—5 mM Tris-HCl, pH 7.5, 600 rpm, F65—20 mM MES, 150 mM NaCl, pH 6.5, 600 rpm; F91—200 mM $KPO_4$, pH 9.1, 600 rpm). The C-terminal truncated α-syn (aa 1–110) was aggregated in 50 mM Tris-HCl 150 mM KCl, pH 7.5, 600 rpm (Bousset et al, 2013; Makky et al, 2016). Fibril concentrations are expressed as the concentration of their constituent α-syn monomers. The aggregated

fibrils were used in subsequent experiments as is (no removal of free monomers or sonication) to avoid any additional clumping or fragmentation of fibrils not specifically caused by fibril conformation or chaperone activities. Free monomer populations after aggregation did not exceed 10% for any of the studied polymorphs, as shown in Fig. 1B.

For sucrose density gradient experiments, pre-aggregated fibrils of the different polymorphs (100 μM) were incubated with AF-555 Succinimidylester (15 μM) in 50 mM $NaHPO_4$, 100 mM NaCl buffer at pH 7.6 for 4 h at RT. Free fluorophore was removed by two times 30 min centrifugation at $435,630 \times g$. Labeling efficiency was determined to be 5–8% by absorbance measurement after denaturation of the fibrils in 5 M GnHCl. Quenching due to high fluorophore density was ruled out by confirming the absence of an increase in fluorescence signal upon denaturation of the fibrils in 4 M GnHCl measured in a CLARIOstar plate-reader (BMG LABTECH, 555 nm) after correction for the different fluorescence intensities of the free AF-555 Succinimidylester in GnHCl compared to in water. 60% AF-555-labeled XG fibrils and AF-555-labeled α-syn monomers are used as controls for fluorophore quenching positive and negative samples, respectively.

## Electron microscopy

EM images were used to visualize differences in fibril polymorphs and samples. Immediately before the sample application, grids (Copper Grids, 300 square, 3.05 mm coated with a carbon film) were glow-discharged for 20 s with 80 mA. Samples were applied at a fibril concentration of 2 μM. Grids were placed on a 10 μL sample drop for 1 min, followed by directly transferring the grid into 3–5 wash steps of 1 mL $H_2O$ drops. The grids were then placed on a drop of 1% uranyl acetate and incubated for 1 min. The excess of uranyl acetate was removed with Whatman paper. Grids were imaged in a transmission electron microscope ZEISS 910 at 80 kV (Carl Zeiss, Oberkochen, Germany) using a slow-scan CCD camera (Albert Tröndle (TRS), Moorenweis, Germany).

## Fibrillar polymorphs fingerprinting by limited proteolysis

As structurally distinct fibrillar α-syn polymorphs exhibit characteristic proteolytic profiles, we subjected each polymorph (1.4 mg/mL equivalent monomer concentration) in PBS at 37 °C to Proteinase K (3.8 μg/mL) (Roche) treatment. Aliquots (10 μL) were removed at different time intervals following the addition of the protease (0, 1, 5, 15, 30, and 60 min) and transferred into Eppendorf tubes containing PMSF (1 μL, 100 mM in ethanol). The samples were dried by speed vacuum and solubilized by the addition of pure hexafluoroisopropanol (HFIP, 30 μL). After overnight incubation at room temperature (RT), the HFIP was evaporated, and the samples were resuspended in Laemmli buffer, heated for 10 min at 80 °C, and processed for Tris-Glycine SDS-PAGE (15%) analysis.

## Thioflavin T (ThT) binding

Differences in ThT binding reactivity were assessed for the different polymorphs, reflecting differences in their fibril core structure. In all, 2 μM fibrils were incubated with 30 μM ThT in 50 mM

HEPES-KOH (pH 7.5), 50 mM KCl, 5 mM MgCl$_2$, 2 mM DTT for 30 min. Fluorescence intensity was measured in a Biotech Omega plate reader at RT (excitation: 440 nm, emission: 480 nm).

## ThT disaggregation assay

The fibril amount in a disaggregation reaction was monitored over time by the ThT signal of the fibrils. In all, 2 µM fibrils were incubated with 4 µM Hsc70, 2 µM DnaJB1, and 0.2 µM Apg2 in reaction buffer (50 mM HEPES-KOH (pH 7.5), 50 mM KCl, 5 mM MgCl$_2$, 2 mM DTT) and 30 µM ThT in the presence of 2 mM ATP and ATP-regeneration system (3 mM PEP and 20 ng/µL pyruvate kinase), as described in Wentink et al (2020). The samples were incubated for 16 h in a 50 µL reaction volume at 30 °C in a BMG Labtech FLUOstar Omega plate reader (Corning assay plate, flat bottom, non-binding surface, black with clear bottom, 96 wells). Measurements were collected at excitation wavelength: 440 nm and emission wavelength: 480 nm.

Background measurements of buffer and inactive chaperone machinery (in the absence of ATP and ATP regeneration system) were subtracted, and all samples were normalized to the fluorescence intensity at timepoint $t = 0$ to facilitate comparison across fibril polymorphs with different ThT reactivities. Experiments were carried out in three biological replicates of different fibril batches. Each biological replicate datapoint is an average of at least two technical replicates.

The twofold increased chaperone concentration experiments were performed with 2 µM fibrils and 8 µM Hsc70, 4 µM DnaJB1, and 0.4 µM Apg2, 2 mM ATP and ATP-regeneration system.

## Supernatant pellet assay

To confirm the results obtained by the ThT disaggregation assay, the disaggregation of fibrils was additionally determined by analyzing the supernatant and pellet fraction before and after incubation with the active chaperone machinery. Fibrils or labeled fibrils (2 µM) were centrifuged for 15 min at $3600 \times g$ after incubation with chaperones (4 µM Hsc70, 2 µM DnaJB1, and 0.2 µM Apg2) in the presence or absence of ATP (2 mM) and ATP-regeneration system for 16 h in reaction buffer. Supernatant and pellet were separated and run on an SDS page gel (Bis-Tris gel 4–20%) followed by immunoblotting (primary antibodies: $\alpha$-syn (SNCA) monoclonal mouse IgG (Sc-12767), $\alpha$-syn (SNCA) (61–95) monoclonal mouse IgG (SM6028), secondary antibody: Alkaline phosphatase coupled anti-mouse-IgG (H + L) (horse)). Membranes were imaged with a LAS-4000 (ImageQuant) instrument and quantified in ImageJ. The released material found in the supernatant was expressed as the fraction of the sum of the pellet and supernatant band intensities.

## Cold denaturation

To measure differences in chemical stability, the different fibrillar polymorphs were incubated on ice. At the indicated time in hours, an aliquot (50 µL) was removed and spun for 30 min at $50,000 \times g$. The supernatant was recovered, denatured with Laemmli buffer, and analyzed by PAGE (15% polyacrylamide), stained with Coomassie blue, and imaged (ChemiDoc MP (BioRad)). To determine the proportion of $\alpha$-syn in the supernatant, the total

amount of $\alpha$-syn is run in parallel. The band intensities were quantified on the ChemiDoc MP (BioRad) used to image the gels and analyzed with ImageLab (version 5.2.1).

## Fluorophore labeling of chaperones

Chaperones mutants (DnaJB1-G194C, Hsc70-C267A-C574A-C603A-T111C) were labeled on free reactive cysteines with Alexa-488 or 594 C5 maleimide. Buffer was exchanged with PD SpinTrap G25 columns into 25 mM HEPES, 150 mM KCl, 5 mM MgCl, and 5% glycerol, and chaperones were incubated for 30 min at 30 °C in the presence of 100 µM TCEP. ATP was added to a final concentration of 5 mM for reactions containing Hsc70 to prevent the labeling of the remaining native cysteine residue C17 in the ATP-binding pocket. An eightfold molar excess fluorophore label was added, and the proteins were incubated for 1 h at RT. Excess fluorophore was removed by two subsequent PD SpinTrap G25 column runs. Labeling efficiency was 90% or higher, based on absorbance measurements of the fluorophore and protein concentrations (Extinction coefficient: Hsc70 33,600 L/mol*cm, DnaJB1 19,033 L/mol*cm).

## Steady-state anisotropy

To determine chaperone affinities for the different fibrils, equilibrium binding was measured by fluorescence anisotropy of labeled DnaJB1-G194C or Hsc70-C267A-C574A-C603A-T111C to fibrils after incubation for 3.5 h at 25 °C (buffer: 50 mM HEPES-KOH (pH 7.5), 50 mM KCl, 5 mM MgCl$_2$, 2 mM DTT), as previously described (Wentink et al, 2020). Fibril titration data of 100 nM AF-594-labeled DnaJB1 mutant G194C or 100 nM AF-488-labeled Hsc70 mutant T111C in the presence of 50 nM DnaJB1 were acquired in CLARIOstar plate-reader (Assay plate, low volume, flat bottom, non-binding surface, black, 384 wells, 10 µL reaction volume) (BMG LABTECH, for AF-488-labeled protein: excitation: 482 nm, emission: 530 nm, dichroic filter: 504 nm; or for AF-594-labeled protein excitation: 590 nm, emission: 675 nm, dichroic filter: 639 nm). Dissociation constants were determined in GraphPad Prism by fitting the data to a model describing equilibrium binding by non-linear regression.

## Sucrose density gradient experiments

Reaction products were separated by sucrose gradient centrifugation. A disaggregation reaction with labeled fibrils (2 µM; 4 µM Hsc70, 2 µM DnaJB1, and 0.2 µM Apg2) was carried out for 20 min, 2 h, and 16 h in the presence of ATP (2 mM) and ATP-regeneration system in reaction buffer at 30 °C. After completion of the reaction time, ATP was depleted with alkaline phosphatase (0.1 U/µL, Roche). As a negative control, alkaline phosphatase was added from timepoint 0 and the reaction incubated for the maximal reaction time of 16 h. Furthermore, samples of untreated fibrils, chaperones only, and monomers were analyzed. Samples (300 µL) were applied on a sucrose gradient (SW40, 10–85% sucrose) and centrifuged for 30 min at 3500 rpm ($217,290 \times g$). Gradients were fractionated by Biocomp Piston Gradient Fractionator™ coupled with a Triax™ flow cell (mCherry). For the quantification of monomer (fractions 0–2), small fragments/oligomers (fractions 3–10), and fibrils (fractions 12–20), fluorescence intensity values of specific fractions

were summed up and divided by the total fluorescence intensity across the gradient. Calculations were performed on baseline-corrected data by subtraction of empty gradient fluorescence intensity values. The plotted data have subsequently been smoothed in Graphpad Prism as a rolling average of 10 datapoints.

## Differential centrifugation of disaggregation reaction products

The products of chaperone-mediated disaggregation were fractionated by distinct centrifugation steps, as shown in Fig. 4C. A disaggregation reaction (1:2:1:0.1 fibrils/Hsc70/DnaJB1/Apg2 in 50 mM HEPES-KOH (pH 7.5), 50 mM KCl, 5 mM MgCl$_2$, 2 mM DTT, 2 mM ATP, and ATP-regeneration system) was incubated for 16 h at 30 °C. After completion of the reaction time, ATP was depleted with alkaline phosphatase (0.1 U/µL, Roche) to quench the reaction. As a negative control, alkaline phosphatase was added from timepoint 0 h and incubated for 16 h ("No disaggregation").

The total sample was divided into three samples. One was not centrifuged to reflect the total disaggregation reaction. The oligomer/small fragment sample was prepared collecting the supernatant after centrifugation at $3600 \times g$ for 15 min to deplete large unprocessed fibrils from the mixture. The monomer sample was depleted of fibrils and oligomers and fibril fragments by centrifugation at $435,630 \times g$ for 30 min.

## In vitro seeding assay

The aggregation behavior of the reaction products was first analyzed in vitro. Aggregation of 20 µM α-syn monomer was induced with 10% seeds in the presence of 30 µM ThT and 0.05% NaN$_2$. Seeds concentrations were expressed as α-syn monomer concentrations. Untreated fibrils, fibrils incubated with chaperones (ATP depletion with alkaline phosphatase (0.1 U/µL, Roche) at timepoint 0 (No disaggregation) or timepoint 16 h (Disaggregation)), and disaggregation reactions fractionated by centrifugation were used as seeds. For the disaggregation reactions, 10 µM fibrils were incubated with 20 µM Hsc70, 10 µM DnaJB1, and 1 µM Apg2 in reaction buffer (50 mM HEPES-KOH (pH 7.5), 50 mM KCl, 5 mM MgCl$_2$, 2 mM DTT) with 2 mM ATP.

The seeded aggregation reactions were monitored under continuous shaking (1000 rpm) for 80 h at 37 °C in a 96-well plate (Corning assay plate, flat bottom, non-binding surface, black with clear bottom) in a Biotech Omega plate reader. Fluorescence intensity measurements were collected at wavelengths of excitation: 440 nm, emission: 480 nm. The initial velocity was calculated by fitting a linear regression over the first 20 min of seeding. An average of the slope of two replicates was determined and normalized to the fibril-only sample within one experiment.

## Seeding assay in cell culture

To confirm in vitro seeding experiments, the seeding capacity of the samples was also tested in cultured cells. Cell culture experiments were performed as previously described (Tittelmeier et al, 2022). Cells were cultured in DMEM (high glucose, GlutaMAX Supplement, pyruvate, 10% FBS, 1× Penicillin–Streptomycin) at 37 °C and 5% CO$_2$. The HEK293T cell line expressing α-syn A53T-YFP was a gift from Marc Diamond, University of Southwestern Texas

(Yamasaki et al, 2019). Cells are routinely tested for mycoplasma contamination using a commercial detection kit (GATC Biotech). Cell line identity was confirmed by evaluating expected phenotypic characteristics, such as homogeneous expression of YFP-tagged α-syn A53T (microscopy, Western blot).

Disaggregation reactions were performed as described under the header "In vitro seeding assay." Cells were seeded with the resulting disaggregation reaction mixtures or fractionated reaction products, depleted of ATP with alkaline phosphatase (0.1 U/µL, Roche) at timepoint 0 (No disaggregation) or timepoint 16 h (Disaggregation) to quench ongoing disaggregation reactions.

The concentration of reaction mixtures or reaction products was adjusted to 2 µM α-syn monomer concentration in Opti-MEM Reduced Serum Medium, GlutaMAX Supplement (Gibco). Pre-incubated Lipofectamine2000 with Opti-MEM (1:20 dilution, 5 min) was added to samples (1:1) for a second incubation step of 20 min. The mixture was then added to cells seeded on coverslips coated with Poly-L lysine (Invitrogen) in 24-well plates in a final concentration of 100 nM. After 48 h, cells were fixed (4% PFA in PBS, 10 min), washed with PBS (3×), and mounted for imaging. Cells were imaged with a confocal microscope, Leica TCS SP8 STED 3X (Leica Microsystem, Germany). Images were processed in ImageJ, and FIJI was used to manually quantify cells and cells with foci. Ten images per condition with 30–100 cells per image were analyzed for each biological replicate. The percentage of cells with foci was calculated and normalized to the percentage of cells with foci in the "No disaggregation" sample to reflect the fold change between conditions. The data was found not to be normally distributed by the Shapiro normality test. Therefore, the significance of all changes was calculated by nonparametric two-way analysis of variance with pairwise comparisons of estimated marginal means with Tukey correction for multiple comparisons.

## Nuclear magnetic resonance (NMR) spectroscopy

### HD exchange NMR experiments
HD exchange experiments were performed as in (Vilar et al, 2008) with some modifications. In all, 40 µM fibrils were centrifuged for 10 min at $3600 \times g$ and resuspended in deuterated buffer. After 3 min of exchange, samples were centrifuged at $20,000 \times g$ for 1 min, the supernatants removed, and the pellets flash-frozen for storage until data acquisition. Fibrillar pellets were resuspended in d6-DMSO + 0.1% TFA 5 min before the start of data acquisition. NMR experiments were recorded on a Bruker 800 MHz spectrometer equipped with a cryogenic probe at 25 °C. All spectra were processed using NMRPipe (Delaglio et al, 1995) and analyzed in Poky (Lee et al, 2021). NMR signal intensities are presented as a ratio of the signal intensity after 3 min of HD exchange relative to a non-exchanged control.

### α-Syn assignment in d6-DMSO
As $^1$H-$^{15}$N chemical shifts in 100% d6-DMSO were incomplete or different compared to previous assignments in other solvents, we acquired triple resonance through-bond experiments at 298 K to obtain backbone resonance assignments using the same spectrometer as for the HD exchange experiments. In addition to standard experiments (Sattler et al, 1999), we recorded an (H)N(COCA)NH experiment to correlate sequential $^{15}$N spins (Bracken et al, 1997). This helped to resolve ambiguities in the $^{13}$C dimension and

allowed a complete backbone assignment (excluding prolines). NMR data were processed using NMRPipe (Delaglio et al, 1995) and analyzed with Cara (http://cara.nmr.ch). The chemical shift assignments have been deposited at the BMRB under the accession code 52999.

## Data availability

The chemical shift assignments of α-synuclein in 100% d6-DMSO have been deposited at the BMRB with the accession code 52999. All data supporting the findings of this study are available within the paper and its Supplemental Material.

The source data of this paper are collected in the following database record: biostudies:S-SCDT-10_1038-S44318-025-00573-3.

## Peer review information

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

## Acknowledgements

The authors would like to thank the Deutsches Krebsforschungszentrum (DKFZ) Electron Microscopy facility and A. Alik for technical support. We are indebted to A. Mogk and M. Mayer for valuable discussions throughout the project. The authors further acknowledge M. Diamond for providing the HEK293 $\alpha$-syn A53T-YFP biosensor cell line. This is an EU Joint Programme–Neurodegenerative Disease Research (JPND) project (PROTEST-70). This project is supported through the following funding organizations under the aegis of JPND—www.jpnd.eu: France, Agence Nationale de la Recherche (ANR, ANR-17-JPCD-0005-01 to RM); Germany, Bundesministerium für Bildung und Forschung (01ED1807A to BB and 01ED1807B to CN-K). This research has been further funded by the Deutsche Forschungsgemeinschaft (DFG, German Research Foundation)—project number 504257241 to BB, a standard grant from the Alzheimer Forschungsinitiative (AFI—17054) to BB, by the Dutch Research Council (NWO, Vidi grant number VI.Vidi.223.172) to ASW, and support from France Parkinson (217681) and EraPerMed DEEPEN-iRBD and Agence Nationale de la Recherche (ANR-22-PERM-0006) to RM. AKB thanks the Novo Nordisk Foundation (grant number NNFSA170028392) and the Lundbeck Foundation (grant number R366-2021-169) for funding. SJ acknowledges the HBIGS graduate school for support.

## Author contributions

**Svenja Jäger**: Conceptualization; Data curation; Formal analysis; Investigation; Visualization; Methodology; Writing—original draft; Writing—review and editing. **Jessica Tittelmeier**: Data curation; Investigation; Writing—review and editing. **Thi Lieu Dang**: Resources; Investigation; Writing—review and editing. **Tracy Bellande**: Resources. **Virginie Redeker**: Resources. **Alexander K Buell**: Conceptualization; Writing—review and editing. **Janosch Hennig**: Formal analysis; Investigation; Writing—review and editing. **Ronald Melki**: Conceptualization; Data curation; Supervision; Funding acquisition; Investigation; Project administration; Writing—review and editing. **Carmen Nussbaum-Krammer**: Conceptualization; Supervision; Funding acquisition; Project administration; Writing—review and editing. **Bernd Bukau**: Conceptualization; Supervision; Funding acquisition; Project administration; Writing—review and editing. **Anne S Wentink**: Conceptualization; Formal analysis; Supervision; Funding acquisition; Writing—original draft; Project administration; Writing—review and editing.

Source data underlying figure panels in this paper may have individual authorship assigned. Where available, figure panel/source data authorship is listed in the following database record: biostudies:S-SCDT-10_1038-S44318-025-00573-3.

## Disclosure and competing interests statement

The authors declare no competing interests.

# Expanded View Figures

**Figure EV1.  High chemical stability of ribbons and F110 polymorphs could explain their resistance to chaperone-mediated disaggregation.** ▶

(A) SDS-PAGE of $\alpha$-syn monomers released to supernatant, separated from fibrillar material by centrifugation, after incubation of fibrils on ice for the indicated times. (B) Fraction of total fibrils depolymerized, as shown in (A), as a function of incubation time on ice. (C) Correlation plot of disaggregation after 16 h in percent (mean ± s.e.m.) and fraction depolymerized with a linear correlation fit (solid line ($r = 0.72$)). (D) ThT disaggregation assay with 1× (XG, blue; F91, red; F65, green; FM, yellow; Ri, gray; F110, purple) or 2× (black) chaperone concentration. Representative graphs of three technical replicates are shown.

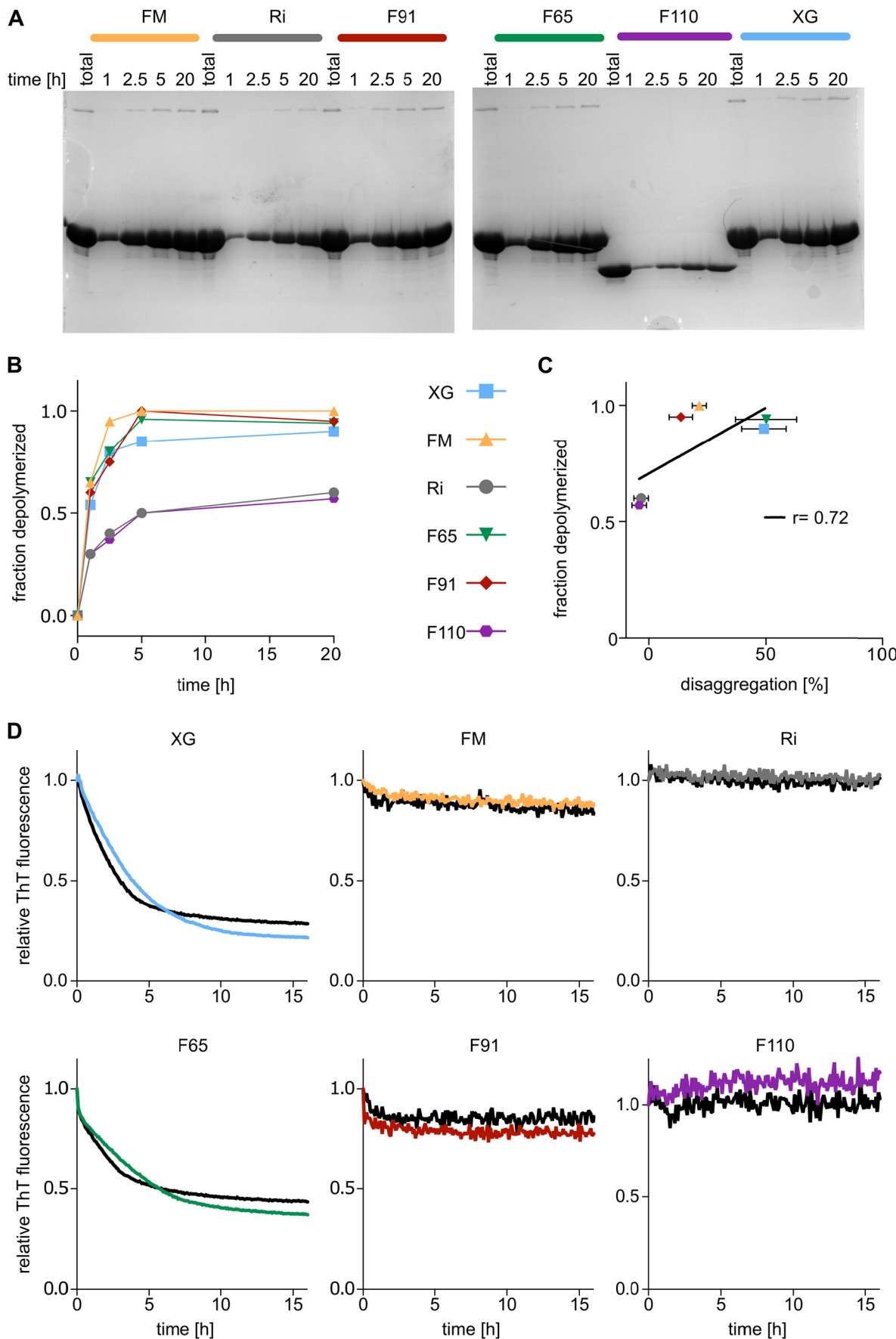

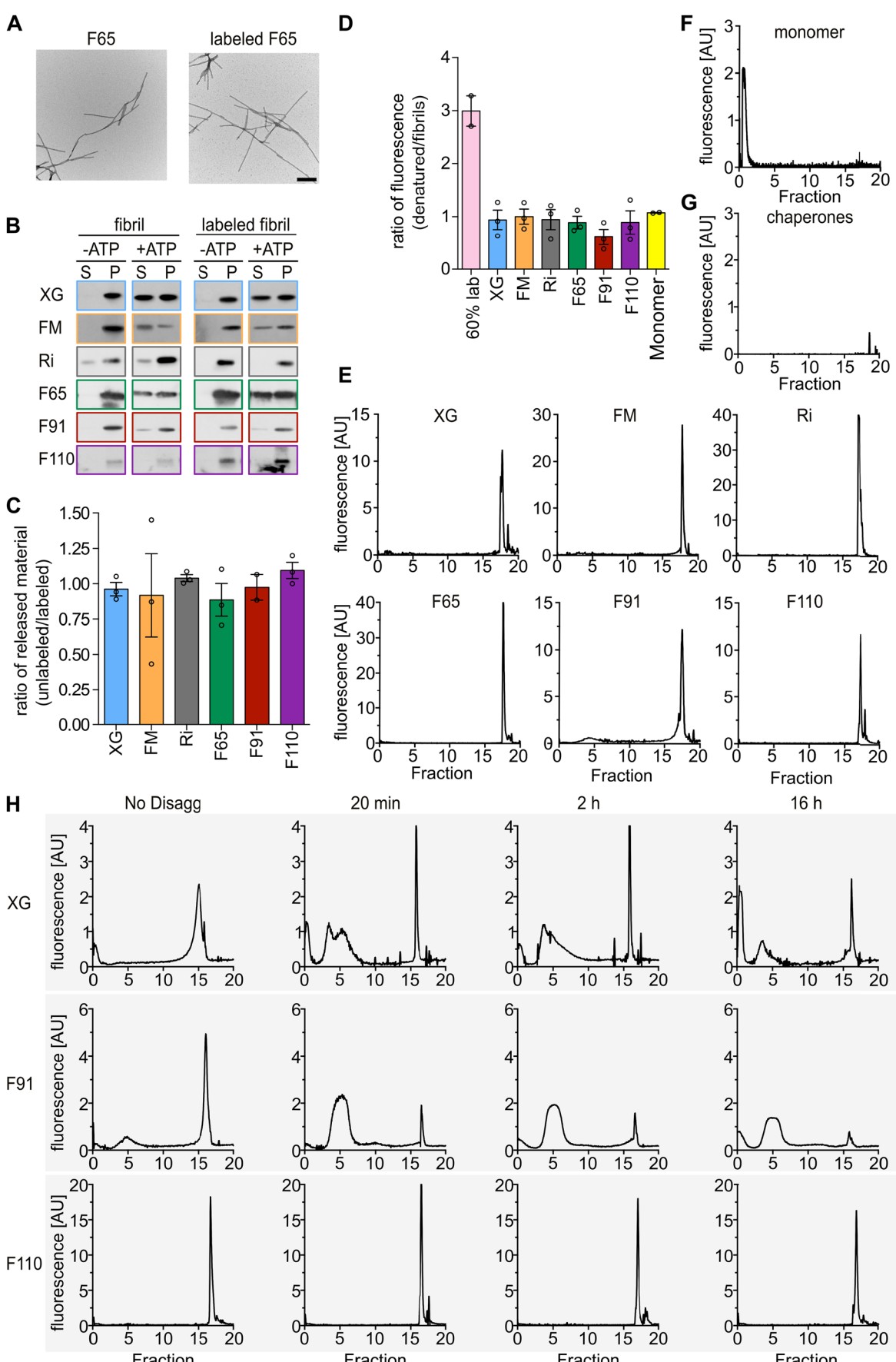

◀

**Figure EV2.** **Fibrillar fragments accumulate to various degrees during chaperone-mediated disaggregation of polymorphs.**

(A) Negative stain EM images of polymorphs F65 unlabeled (left) and AF555-labeled (right) (scale bar 200 nm). (B) Representative western blot images of unlabeled (left) and AF555-labeled polymorphs (right) incubated with the chaperone machinery in the presence (+ATP) and absence of ATP (−ATP). (C) Ratio of released α-syn protein of AF555-labeled and unlabeled fibrils of all polymorphs. Quantification of protein in the supernatant of the total (P + S) after 16 h disaggregation by the active chaperone machinery in (B) (analyzed with ImageJ). Data are mean ± s.e.m., $n = 3$, except polymorph F91 where $n = 2$. (D) Ratio of normalized fluorescence of denatured AF555-labeled fibrils/monomer in GnHCl compared to labeled fibrils/monomer in buffer. Data are mean ± s.e.m., $n = 3$, except 60% labeled fibrils and monomer where $n = 2$. (E, F) Sucrose density gradient (10–85%) profile of AF555-labeled monomers (E) and chaperones (Hsc70, DnaJB1, Apg2) only (F). Sucrose gradient of labeled fibrils (XG, FM, Ri, F65, F91, F110) is shown in (G). (H) Representative sucrose gradient (10–85%) profile of a polymorph XG, F65, F91, and F110 after incubation of AF555-labeled fibrils with the active chaperone machinery (Hsc70, DnaJB1, Apg2, + ATP) for 20 min, 2 h, 16 h, and 16 h incubation with the inactive machinery (−ATP, No Disaggregation).

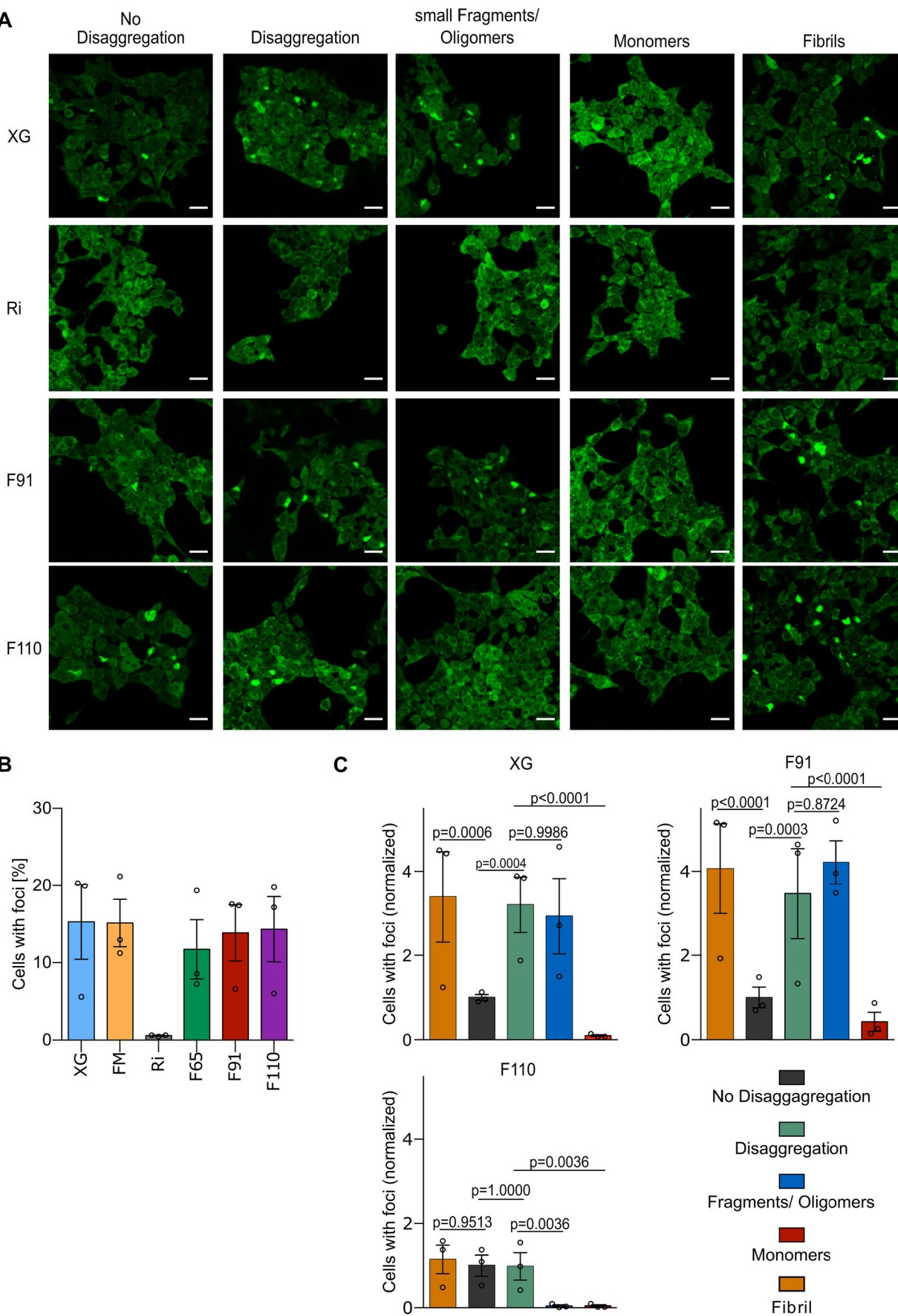

**Figure EV3. Disaggregation reaction products trigger foci formation in the human cell model.**

(A) Representative fluorescence intensity microscopic images of HEK293T cells stably expressing α-syn A53T-YFP seeded and treated with fibril preparations of polymorph XG, Ri, F91, and F110. The cells were exposed to fibrils only, fibrils incubated with chaperones in the absence (No Disaggregation) and presence of ATP (Disaggregation), as well as small fragments/oligomer and monomer fractions of the disaggregation reaction separated by centrifugation as described in Fig. 4C. Scale bar corresponds to 20 μm. (B) Percentage of cells with foci for the individual polymorphs ($n = 3$). (C) Quantification of the percentage of cells with foci normalized to the No disaggregation sample (No disaggregation, black; Disaggregation, green; small fragments/oligomers, blue; monomers, red; fibrils, orange). The Ri polymorph did not induce aggregation in the used reporter cell system (in (B)) and was therefore not further analyzed. Data are mean ± s.e.m., $n = 3$. Statistical analysis was performed by nonparametric two-way ANOVA with pairwise comparisons of estimated marginal means with Tukey correction for multiple comparisons. Exact p-values are indicated in the figure.

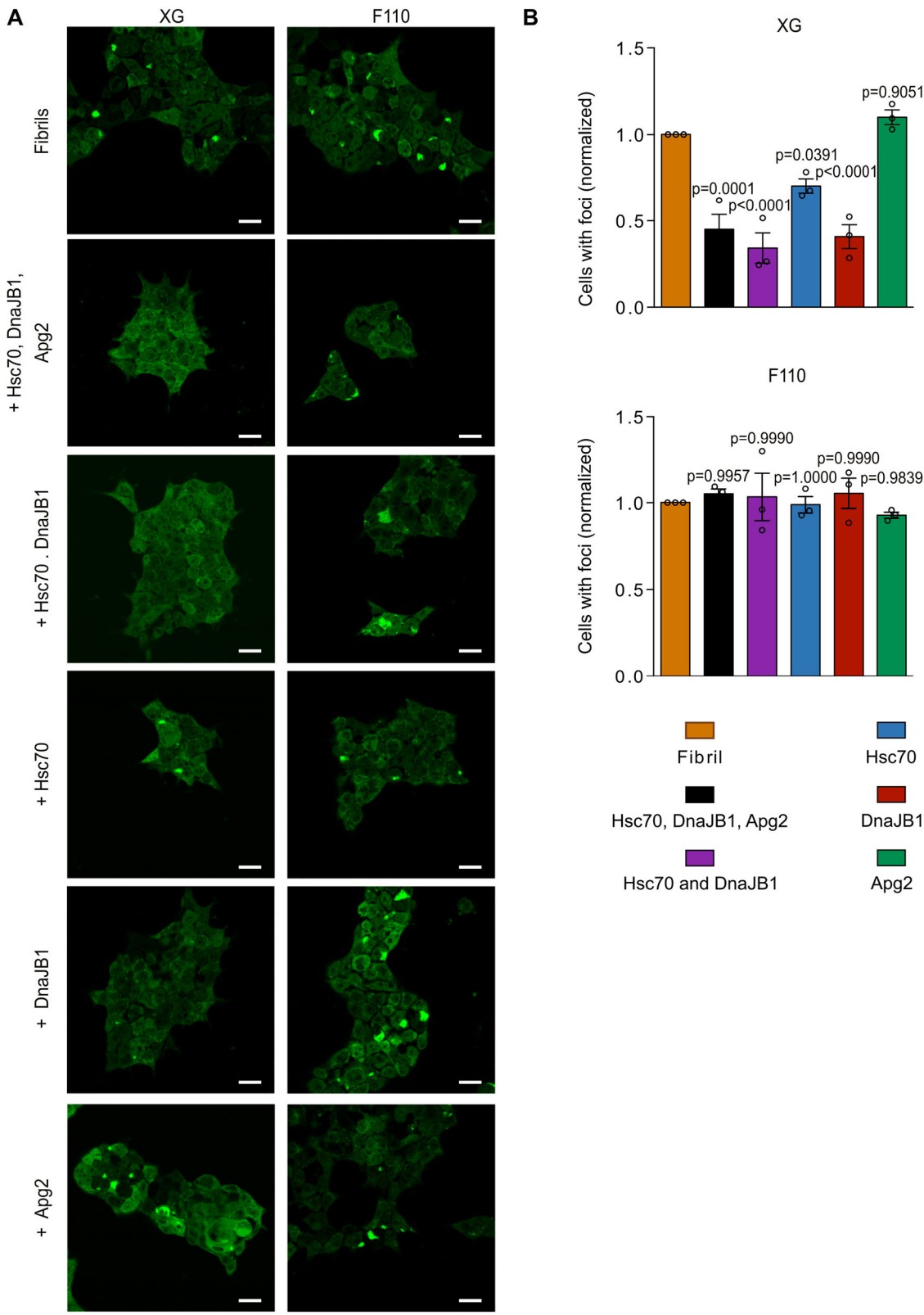

◄ **Figure EV4. Pre-incubation of α-syn fibrils with DNAJB1 reduces foci formation.**

(A) Representative fluorescence intensity microscopic images of HEK293T cells stably expressing α-syn A53T-YFP seeded with fibrils (polymorphs XG and F110) incubated with different chaperone combinations; fibrils without chaperones as a control, fibrils with the whole chaperone machinery (Hsc70, DnaJB1, Apg2), Hsc70 and DnaJB1 and Hsc70, DnaJB1 and Apg2 individually. Scale bar corresponds to 20 μm. (B) Quantification of the percentage of cells with foci normalized to fibril-only conditions (XG, top; F110, bottom). Data are mean ± s.e.m., $n = 3$. Statistical analysis was performed by nonparametric two-way ANOVA with pairwise comparisons of estimated marginal means with Tukey correction for multiple comparisons. Exact p-values are indicated in the figure.

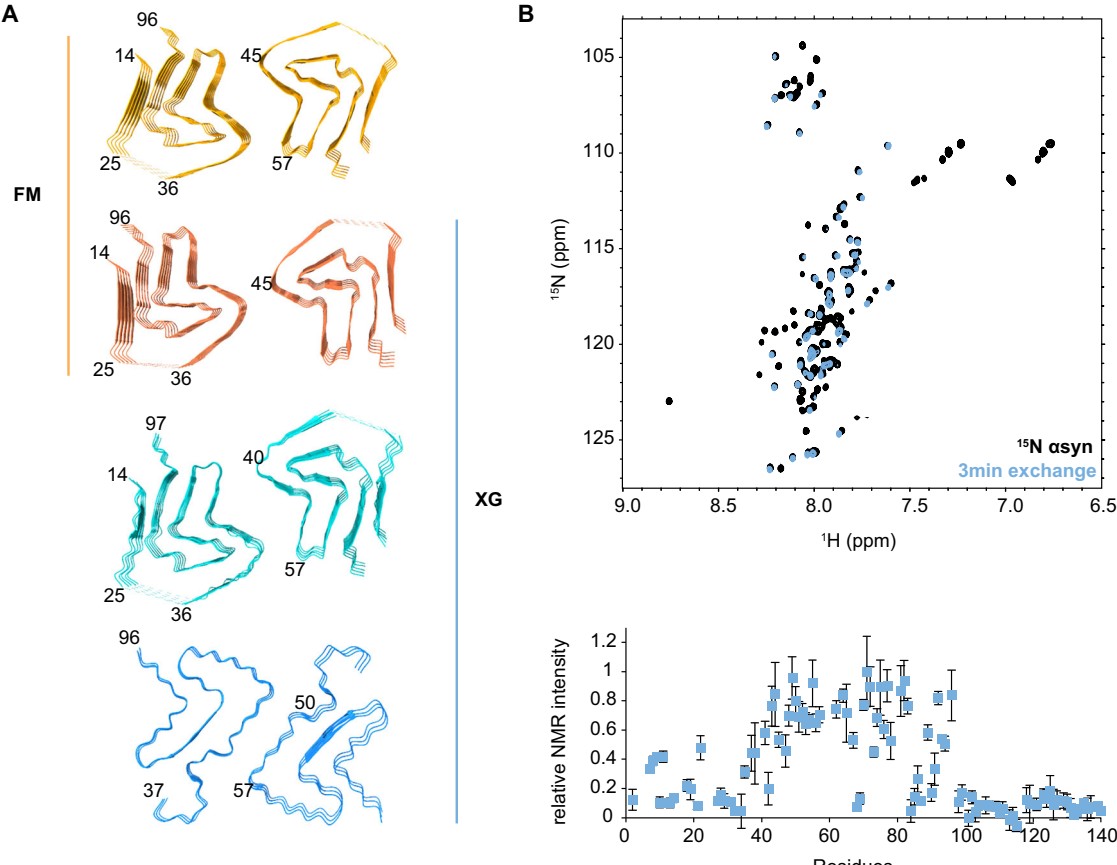

**Figure EV5.  Structural comparison of α-syn polymorphs FM and XG.**

(**A**) Cryo-electron microscopy-derived structural models of different fibril conformations generated under the FM (PDB: 6SSX and 6SST) (Guerrero-Ferreira et al, 2019) and XG (PDB: 8RRR and 8RQM) (Monistrol et al, 2025) aggregation conditions. (**B**) Hydrogen–deuterium exchange of the XG polymorph followed by NMR spectroscopy. NMR resonance intensities after 3 min of HD exchange in deuterated buffer are plotted relative to their non-exchange intensities as a function of the *a*-syn sequence. The N- and C-terminal sequences exchange rapidly and are thus mostly disordered, while residues 38–96 make up the slow-exchanging fibril core. Data are mean ± s.d., n = 3.

