## [Peer Review File · The EMBO Journal]

Structural polymorphism of α -synuclein fibrils alters pathway of Hsc70 mediated disaggregation

Svenja Jäger, Jessica Tittelmeier, Thi Lieu Dang, Tracy Bellande, Virginie Redeker, Alexander Buell, Janosch Hennig, Ronald Melki, Carmen Nussbaum-Krammer, Bernd Bukau, and Anne Wentink

Corresponding author(s): Anne Wentink (a.s.wentink@lic.leidenuniv.nl) , Bernd Bukau (bukau@zmbh.uni-heidelberg.de), Carmen Nussbaum-Krammer (carmen.nussbaum@med.uni-muenchen.de), Ronald Melki (ronald.melki@cnrs.fr)

Review Timeline:

Submission Date:	1st Dec 24
Editorial Decision:	10th Feb 25
Revision Received:	11th May 25
Editorial Decision:	14th Jul 25
Revision Received:	29th Jul 25
Accepted:	27th Aug 25

Editor: Cornelius Schneider

Transaction Report:

Dear Dr. Wentink,

Thank you for submitting your manuscript for consideration by the EMBO Journal. It has now been seen by three referees whose comments are shown below. Thank you also for the productive discussion.

Given the referees' positive recommendations, I would like to invite you to submit a revised version of the manuscript, addressing the comments of all three reviewers as discussed in the pre-decision consultation. I should add that it is EMBO Journal policy to allow only a single round of revision, and acceptance of your manuscript will therefore depend on the completeness of your responses in this revised version.

Thank you for the opportunity to consider your work for publication. I look forward to your revision.

Yours sincerely,

Cornelius Schneider, PhD
Editor
The EMBO Journal
c.schneider@embojournal.org

Please remember: Digital image enhancement is acceptable practice, as long as it accurately represents the original data and conforms to community standards. If a figure has been subjected to significant electronic manipulation, this must be noted in the

figure legend or in the 'Materials and Methods' section. The editors reserve the right to request original versions of figures and the original images that were used to assemble the figure.

We realize that it is difficult to revise to a specific deadline. In the interest of protecting the conceptual advance provided by the work, we recommend a revision within 3 months (11th May 2025). Please discuss the revision progress ahead of this time with the editor if you require more time to complete the revisions. Use the link below to submit your revision:

Referee #1:

This manuscript by Jäger et al. explores how the human Hsc70 chaperone machinery disaggregates structurally distinct polymorphs of α -synuclein fibrils. Using in vitro aggregated conformational polymorphs, the authors showed that structural variations influence both the efficiency of fibril clearance and the disaggregation pathway preferred by the Hsc70 machinery. Additionally, they find that polymorphs resistant to chaperone activity are more prone to fragmentation, producing highly seeding-competent species that exacerbate protein aggregation.

Overall, the study highlights that fibril disaggregation efficiency by the Hsc70 machinery is governed by a combination of chaperone binding and fibril stability, with the generation of seeding-competent species as a byproduct. However, the conclusions are not fully substantiated by the presented data, and some findings appear to reiterate those from prior studies (DOI: 10.1074/jbc.RA120.013478, DOI: 10.1101/2024.12.19.629136).

In light of these concerns, I am unable to recommend this manuscript for EMBO J. My specific concerns are listed below.

Major concerns:

1. The work focuses on how different structural polymorphic fibrils behave differently upon engaging with Hsc70. However, the structural characterization of the fibrils studied in this work is very preliminary. The authors should provide a more comprehensive analysis of the polymorph structures and establish clear correlations between their structural features and disaggregation activity. For example, they could investigate whether the N- and C-termini of the fibrils are incorporated into the fibril core, potentially hindering chaperone binding, as chaperones typically bind to these regions. Additionally, the relationship between structural variations and the thermodynamic stability of the fibrils should be explored in greater detail.
2. The polymorphism-regulated disaggregation mechanism requires further exploration, not just a declaration. For example, if disaggregation occurs from the ends of specific conformational fibrils, what makes the chaperone more likely to bind to the ends of specific α -syn polymorph fibrils?
3. The authors claim that chaperone affinity may determine the dominant disaggregation pathway. However, the statement that high affinity leads to depolymerization via binding at the fibril ends, while low affinity leads to fragmentation through binding on the fibril surface, lacks logical support. Given that the exposed fibril ends are far fewer than the exposed surfaces, if the chaperone preferentially binds to the surface, its affinity for this binding mode should not be lower than its affinity for binding at the ends.
4. Hsc70 is primarily responsible for fibril disaggregation, however, in this study, the correlation between Hsc70 binding and disaggregation efficiency is relatively low. This observation requires further investigation and a more detailed explanation.
5. In Figure 3B, the trend observed in F65 is not sufficiently distinct to support the claim that disaggregation starts from the ends, which may impact the key conclusions of the manuscript.

Minor concerns:

1. The electron microscopy images provided by the authors show that some fibrils (F91, F110) are clearly bundled together, while others are not. This phenomenon could influence chaperone binding and disaggregation. The authors should acknowledge this factor instead of focusing solely on structural differences.
2. In this study, different polymorphs were obtained under various preparation conditions, including varying salt concentrations and pH levels. However, salt concentration seems to significantly influence chaperone binding to α -syn fibrils (DOI: 10.1101/2024.01.25.577078). The authors should take this factor into account in their experimental design.
3. The authors propose that fibril polymorphism explains the conflicting observations of depolymerization and fragmentation in previous studies. However, prior research using the same preparation conditions for FM fibrils as in this study reported disaggregation through depolymerization (DOI: 10.1038/s41467-021-25966-w), which contradicts the conclusions presented in this paper.
4. The authors state that the initial aggregation rate of α -syn monomers seeded with the F65 low-speed centrifugation fraction is reduced by 25%, indicating that 75% of the starting material has been processed into smaller fibrils or oligomers in the soluble fraction. However, this consistency seems unlikely because, after the disaggregation reaction, in addition to 25% of the fibrils being removed by centrifugation, some fibrils should have been converted into monomers with no seeding ability. Therefore, the

initial seeding rate of the low-speed centrifugation fraction should be lower than 75%, not exactly 75%.

Referee #2:

Highly important topic: Thanks to the cryoEM revolution, we now are aware of amyloid polymorphism and of its apparent importance in disease: certain polymorphs appear to associate to certain pathologies.

The key question now is: how does the polymorph affect the effect of amyloids on the cell and tissue? Also, what mechanisms lead to polymorph selection in disease?

The current manuscript by Wentink and colleagues addresses an interesting angle in this area, namely how differential interactions with Hsp70 chaperones (and cochaps) with polymorphs of alphasynuclein leads to different processing of asyn amyloids by the chaperone. This capitalises on previous work of Ronald Melki on the controlled production of various polymorphs and of the Nussbau-Krammer and Bukau labs on the disassembly of amyloids by Hsp70. With Alex Buell thrown in for good (biophysical) measure, quite the cast;-)

Perhaps not surprisingly to a structural biologist (but still important to show), the atomic detail of the polymorphs appear to matter: they affect how and how well Hsp70/DNJB1 can engage the fibrils. In some cases this can lead to efficient clearance, in some cases it actually produces seeds that promote further aggregation (this was already known from propagation studies of the yeast prion sup35 many years ago, it would be good to cite those).

I am enthusiastic about this paper, but the part on seeding is a little unclear to me (Figures 4 and 5), and I find this finding important

- seeding is done with intact fibrils? The standard in the field is to sonicate fibrils to get maximal seeding efficiency. How does that look on this scale (it should be much stronger)
- These curves don't look like typical seeding curves: seeding is usually defined as the disappearance of a lag phase and as a quantification, you would show the lag phase in the seeded and unseeded conditions. Here there is no lag phase to begin with, so it looks more like an acceleration of the aggregation reaction, not as the bypassing of the rate-limiting nucleation step
- It is unclear to me how the 'seeding (normalised)' values are computed here
- The y-axis "ThT signal", specify how it was normalised (and show unnormalised curves in supplement),
- in the quantification of the biosensor microscopy images, the y axis is labelled as 'cells with foci'. Is that fraction or percentage? is the transfection efficiency somehow corrected for or not?
-

Is there any high resolution cryoEM data on these polymorphs in the public domain? In this case they should be shown and an attempt should be made to interpret them in light of these data.

I think the main limitation of the study is by focussing on 1 particular Hsp70/DNAJ, the authors may be overestimating the stability of certain fibrils in the face of a chaperone attack: perhaps in cellular conditions, other diversity of the chaperones deals with this issue and in the end all polymorphs are destroyed. I think this should be clearly included as a caveat.

- the time scales of disaggregation and seeding appear to be quite different, it would be interesting to discuss

Referee #3:

This work explores the hypothesis that the structure (polymorph) of synuclein fibrils might alter their susceptibility to disaggregation by the Hsp70-mediated system. The authors create recombinant synuclein fibrils under different buffer conditions (and use a C-terminal truncation) to alter the polymorph, using partial proteolysis, negative stain TEM and denaturation curves to document the differences between these structures. Then, they treat with the Hsp70 system (either w/ or w/o ATP), using fluorescence labelling, sucrose gradients and ThT assays to quantify impacts on fibril integrity and the relative size of the products. Finally, they use a biosensor assay in cells to measure relative seeding capacity of these products. The work is carefully performed and clearly written. The methods are key to judging this work and they are, in large part, clearly stated. The statistical methods are also clearly described.

From the experiments, the authors show that multiple factors, including intrinsic fibril stability and overall architecture are indeed important in determining whether the Hsp70 system will dismantle synuclein fibrils to low molecules mass (e.g. monomer) structures. For example, the most stable fibril (Ri) is largely impervious to the Hsp70 system. This work adds important details to our understanding of the human disaggregase - with the major strength being the conclusion that fibril conformation impacts

disaggregase efficiency. This reviewer only had a couple of points for consideration, largely focused on enhancing clarity of the presentation.

1. While the authors focus on differences in fibril structure, their conclusions formally require that the total amount or length of the fibrils is relatively uniform. For example, under extreme conditions in which comparison are made between samples that contain sparse fibrils vs. samples with numerous fibrils - then one might be "tricked" into thinking that the resulting differences in disaggregase activity are due to the relative structures of the fibrils when they are actually due to molar ratio. While it is challenging to convincingly document fibril concentration, some additional experimental details would be helpful. The methods indicate that the fibril concentrations are based on monomer concentration. So, did 100 percent of the monomer progress into the fibril form? This efficacy value could be quantified using the sucrose gradient? Or were the fibrils first purified away from unreacted monomer?

2. It is not entirely clear that these experiments can fully differentiate between the steps of fibril fragmentation vs monomer extrusion. The experiments in Fig 3 are compelling to suggest that fibrils proceed to oligomers and then monomers (for FM), but there does not appear to be full conservation of mass (fluorescence) in the plots. Under those conditions, a small amount of (slow) monomer release directly from fibril could still be possible. There might be ways around this (e.g. lots of kinetic TEM, single particle tracking microscopy, plus separating the bands on gels), but ultimately it might be best to soften the language around the relative contributions of fibril fragmentation and monomer extrusion. To be clear, this reviewer thinks that the authors conclusion is likely accurate, but the experiments are not entirely convincing.

Minor:

1. It would be useful to include the volumes for the ThT assays, along with other details such as the microtiter plate type.
2. The buffer conditions for the synuclein fibril preparation are really important for this work and it might be worth putting the conditions in a Table with the citations (which could be easier to read/compare) than in text.

Dear Editor,

We would like to thank all reviewers for their time and constructive feedback. Please find our point-by-point response below:

Referee #1 (Report for Author)

This manuscript by Jäger et al. explores how the human Hsc70 chaperone machinery disaggregates structurally distinct polymorphs of α -synuclein fibrils. Using in vitro aggregated conformational polymorphs, the authors showed that structural variations influence both the efficiency of fibril clearance and the disaggregation pathway preferred by the Hsc70 machinery. Additionally, they find that polymorphs resistant to chaperone activity are more prone to fragmentation, producing highly seeding-competent species that exacerbate protein aggregation.

Overall, the study highlights that fibril disaggregation efficiency by the Hsc70 machinery is governed by a combination of chaperone binding and fibril stability, with the generation of seeding-competent species as a byproduct. However, the conclusions are not fully substantiated by the presented data, and some findings appear to reiterate those from prior studies (DOI: 10.1074/jbc.RA120.013478, DOI: 10.1101/2024.12.19.629136).

In light of these concerns, I am unable to recommend this manuscript for EMBO J. My specific concerns are listed below.

We thank the reviewer for their honest feedback. We however disagree that our study simply reiterates the cited prior studies:

Nachmann et al (DOI: 10.1074/jbc.RA120.013478) demonstrate for the protein Tau that the Hsp70 based disaggregase can tackle amyloid fibrils of different isoforms and show that a low MW species is released that can seed aggregation. These isoforms are splicing variants and thus have different primary sequences. The current study instead deals with polymorphs of alpha-synuclein fibrils, hence different fibrillar conformations of the same WT protein sequence which display identical binding sequences that, we show, nevertheless are differentially recognised by the Hsp70 chaperone machinery. We provide biochemical data to conclude that the different sensitivities of the fibril conformations correlate well with the binding of DNAJB1, the targeting factor of Hsp70. Through a detailed characterization of reaction products, we show that intermediate species released during disaggregation (rather than low MW for tau) are seeding competent and that this is most pronounced for polymorphs more susceptible to accumulating fragments during disaggregation. The current work thus highlights a different angle of structural polymorphism among amyloid fibrils (not driven by sequence differences) and provides more mechanistic insight in the molecular basis of the observed differences.

Fricke et al (DOI: 10.1101/2024.12.19.629136) is a preprint of a parallel study, including 4 of the same authors, that focusses more in depth on the role of thermodynamic stability in polymorphism and the consequences for disaggregation. Note that the publication date of the Fricke study is 2 weeks after the preprint deposit and EMBO submission of our manuscript and therefore cannot claim precedence over the current work. We have chosen to separate these two studies to keep the emphasis here on the differential recognition by chaperones and

the functional consequences thereof and do full justice to the impact of thermodynamic stability and its time evolution in a separate study whose extensive experimental design was specifically geared towards addressing this question.

Major concerns:

1. The work focuses on how different structural polymorphic fibrils behave differently upon engaging with Hsc70. However, the structural characterization of the fibrils studied in this work is very preliminary. The authors should provide a more comprehensive analysis of the polymorph structures and establish clear correlations between their structural features and disaggregation activity. For example, they could investigate whether the N- and C-termini of the fibrils are incorporated into the fibril core, potentially hindering chaperone binding, as chaperones typically bind to these regions. Additionally, the relationship between structural variations and the thermodynamic stability of the fibrils should be explored in greater detail.

The FM, Ri, F91 fibril polymorphs used in this study have been extensively characterized from a structural, biochemical and biophysical perspective in previous work (Bousset et al 2013; Landureau et al 2021; Verasdonck et al 2016; Makky et al 2016; Peelaerts et al 2015). We have modified the final figure 6 to include a more detailed summary of the available structural data of the fibril polymorphs (Fig. 6B).

Cryo-EM structures of the FM and (chaperone-bound) XG polymorphs have been published and help to define the fibril core boundaries (New Fig. EV 5A). In addition, we have performed HD exchange NMR spectroscopy of the less well characterized XG polymorph to understand which part of the sequence remains highly dynamic in the amyloid state (Fig. EV 5B). We find that the C-terminus rapidly exchanged, indicating disorder, while regions between positions 35 and 96 are more highly protected and thus are likely to be part of the amyloid core. The N-terminus is also mostly disordered but some regions near the extreme N-terminus are more protected relative to the C-terminus suggesting some degree of (transient) structure or compactness.

The engagement of the N- and C-termini of alpha-synuclein polymorphs has been also assessed in previous solid state NMR structural studies. The C-terminus has been found to never be involved in the fibrillar scaffold. The N-terminus in contrast can be part of the amyloid core (ribbons) or mostly unstructured (F91) as shown in Fig 6B. Structural variations in these flexible regions have also been assessed by differential binding of small ligands (DOI: 10.1021/acchemneuro.4c00178), aptamers (DOI: 10.1093/nar/gkae544), and HDX (see above).

Generally, the presence of a disordered C-terminus is a prerequisite for DNAJB1 binding that all polymorphs, except F110, meet. The detected difference in binding affinity can therefore not readily be derived from the available, or any additional, high resolution structural information. It is likely that nonspecific interactions with the fibril surface as well as the relative spacing between disordered tails (due to fibril twist) contribute to the observed differences, as avidity driven binding modes can be highly sensitive to such subtle differences.

The observed differences in N-terminal structure and dynamics suggests (partial/ transient) structure in this region may interfere with productive Hsp70 binding when the N-terminus is

more ordered and thus requires a different Hsp70 binding mode than the flexible tail equivalents. In agreement, we observe that a higher degree of protection from HD exchange of the N-terminus correlates with a decreased sensitivity to disaggregation by the Hsc70 machinery.

Finally, we made a strategic decision to separate the consideration of thermodynamic stability and chaperone binding to avoid overcrowding the manuscript with data and keep the focus in this study on the specific activity of chaperones, the resulting differences in disaggregation pathways and the physiological consequences thereof, e.g. for seeding of amyloid formation. The relationship between structural variations and the thermodynamic stability is documented in more detail in the parallel study by Fricke et al (DOI: 10.1101/2024.12.19.629136) where the focus is on the biophysical properties of the polymorphs.

2. The polymorphism-regulated disaggregation mechanism requires further exploration, not just a declaration. For example, if disaggregation occurs from the ends of specific conformational fibrils, what makes the chaperone more likely to bind to the ends of specific α -syn polymorph fibrils?

This concern seems to be rooted, at least in part, in a misunderstanding. We do not intend to claim that chaperones bind preferentially to fibril ends in those instances where significant monomers are released at early timepoints in the disaggregation reaction. The only study we are aware of that has looked at the positional preference of the Hsp70 chaperone along the fibril axis (Beton et al EMBO J 2022) demonstrated that Hsp70 accumulated preferentially at one end of the fibril relative to the other end. It however did not quantify the enrichment between fibril ends and along the fibril and indeed this same study reported instances of fragmentation, alongside processive bursts of depolymerisation for the XG polymorph, suggesting that the observed positional bias is not incompatible with fragmentation events. Lacking experimental data to support a particular positional preference of chaperones, we defer to a more minimal model, where the flexible tails of all alpha-synuclein monomers are recognized in the same manner irrespective of their relative position in the fibril. In this case, chaperones would bind the fibril surface at random; however the stochastic assembly of the disaggregase machinery near the fibril ends would be more productive as monomers here are easier to extract than in the middle of the fibril. How frequent chaperones bind near the fibril ends, at random, will depend on chaperone affinity (in particular that of DNAJB1 as targeting factor for Hsp70). Whether this binding indeed results in disaggregation by depolymerisation will depend on the stability of the monomers at the fibril ends. Perhaps counterintuitively, low affinity of DNAJB1 for fibrils could hypothetically be favourable to fragmentation, as fewer JDPs bound per fibril would recruit catalytically from a similar pool of Hsp70s, thus resulting in relatively more localized clustering of Hsp70 (fewer clusters that contain more Hsp70s each), and therefore may generate of a stronger entropic pulling force that would facilitate fragmentation (Wentink et al, Nature, 2020). The positional dependence of chaperones and the composition of Hsc70 clusters that are conducive to fragmentation or depolymerization is an intriguing question for follow up studies but cannot be address within the current experimental design. In the text we now therefor simply state: “The observation that DNAJB1 affinity correlates strongest with disaggregation efficiency is consistent with its critical role in the functional recruitment of Hsc70 into higher order assemblies (Faust et al, 2020), that are key to generating the forces required for amyloid disaggregation (Wentink et al, 2020; De Los Rios et al, 2006;

Goloubinoff & De Los Rios, 2007). ” but do not speculate further as to what features of these Hsc70 assemblies may specify the preferred disaggregation pathway.

3. The authors claim that chaperone affinity may determine the dominant disaggregation pathway. However, the statement that high affinity leads to depolymerization via binding at the fibril ends, while low affinity leads to fragmentation through binding on the fibril surface, lacks logical support. Given that the exposed fibril ends are far fewer than the exposed surfaces, if the chaperone preferentially binds to the surface, its affinity for this binding mode should not be lower than its affinity for binding at the ends.

As stated above, we do not assume chaperones preferentially bind the fibril ends versus fibril surface, or that such preference would correlate with a preferred disaggregation pathway. On the basis of our experiments, we cannot distinguish this, as noted by the reviewer, and we do not believe that this is necessary to assume to interpret our data. See our response to point 2 for more details. Of course, without additional (single molecule) kinetic experiments and structural work probing the architecture of Hsp70 under different conditions, which would be an entirely new study, the potential bias towards fragmentation at low DNAJB1 affinities, or concentrations, remains a model that will require further validation.

4. Hsc70 is primarily responsible for fibril disaggregation, however, in this study, the correlation between Hsc70 binding and disaggregation efficiency is relatively low. This observation requires further investigation and a more detailed explanation.

In Wentink et al, 2020 and Faust et al, 2020, we demonstrated that simply recruiting Hsp70 to the fibril is not enough to enable disaggregation, but rather the higher order spatial organization of Hsp70 molecules (clustering) is key to disaggregation. For example, the J-domain protein DNAJA2 did not support disaggregation, despite increasing Hsp70 affinity for the fibrils. Thus, we do believe that our observation that DNAJB1 binding correlates more strongly with disaggregation than Hsp70 binding is consistent with previous reports as DNAJB1 would be essential to recruit Hsp70 into functional complexes. We have added this explanation to the section “Differences in polymorph structures perturb interactions with the chaperone machinery“.

5. In Figure 3B, the trend observed in F65 is not sufficiently distinct to support the claim that disaggregation starts from the ends, which may impact the key conclusions of the manuscript.

We thank the reviewer for pointing out that the F65 trend shown in Figure 3B is difficult to interpret. In response, we have revisited the raw data for the F65 sucrose density gradient experiments. The three traces shown in Fig 3B correspond to representative curves picked amongst 2-3 technical replicates for 3 different fibril preparations (batches). We here show the 3 technical replicates for the batch indicated by a triangle in the original figure, which shows somewhat inconsistent behaviour compared to the other 2 batches.

We originally selected to display the first repeat due to its higher signal to noise (higher labelling efficiency) but find it shows somewhat delayed disaggregation compared to the other two replicates, and the other F65 batches. We now have replaced this data set with repeat 2 (green), which shows behaviour intermediate to both other technical replicates and

does not suffer from the drifting baseline of repeat 3 and is thus more representative of general behaviour of this fibril batch.

We believe the shown data now more accurately reflects the general behaviour of the F65 polymorph, and the individual biological replicates. While the overall experimental variation for this polymorph is indeed greater than for the others, the data consistently shows rapid processing of the fibrils with efficient conversion of the oligomer/fragment fraction to monomers, well beyond what is observed for any of the other polymorphs. We thus believe that our conclusion, that depolymerization is most efficient for this polymorph, and that its pathway of disaggregation is distinct, is supported by the data.

Minor concerns:

1. The electron microscopy images provided by the authors show that some fibrils (F91, F110) are clearly bundled together, while others are not. This phenomenon could influence chaperone binding and disaggregation. The authors should acknowledge this factor instead of focusing solely on structural differences.

Higher order assemblies were indeed regularly observed for the F110 conformation, but not the F91 conformation as can also be seen in previous studies (Makky et al, 2016; Shrivastava et al 2022; Landureau et al, 2021 and others). We have swapped the representative image for F91 in Fig 1 to avoid giving the impression that bundling is a standard feature of the F91 polymorph. We concur that the observed bundling would make disaggregation of F110 fibrils potentially more challenging but we would also argue that this would be a relevant conformation specific property that modulates the disaggregation activity. We have now acknowledged this as a potentially relevant factor in the discussion.

2. In this study, different polymorphs were obtained under various preparation conditions, including varying salt concentrations and pH levels. However, salt concentration seems to significantly influence chaperone binding to α -syn fibrils (DOI: 10.1101/2024.01.25.577078). The authors should take this factor into account in their experimental design.

All disaggregation assays have been performed in the same buffer, with fibril stocks diluted a minimum of 100-fold from the conditions of their original formation. Even in experiments where high fibril concentrations were used (chaperone titration experiments) all stock were diluted at least 10-fold (except XG where the dilution factor is 5 at 40 μ M in the Hsc70 titration). We thus do not expect that the resulting minor differences in salt concentrations (a maximum variation between 45-60mM KCl at the highest fibril concentrations) would meaningfully alter the apparent dissociation constant measured to such an extent to affect our conclusions.

Note that we have intentionally avoided buffer exchanging the fibrils before mixing with chaperones as in our experience, sedimentation of the fibrils results in significant higher order assembly that can be difficult to dissolve without sonication. Given that our study specifically investigates fragmentation, we decided to avoid where possible any intervention that generates a breakdown of fibrils before chaperone treatment. This was thus a conscious trade-off in the design of these experiments.

3. The authors propose that fibril polymorphism explains the conflicting observations of depolymerization and fragmentation in previous studies. However, prior research using the same preparation conditions for FM fibrils as in this study reported disaggregation through depolymerization (DOI: 10.1038/s41467-021-25966-w), which contradicts the conclusions presented in this paper.

The protocol followed for the aggregation by the cited work diverges from our protocol on some key points: 10% labelled monomers were present during aggregation (while in the instances we used fluorescently labelled fibrils, these were labelled post-aggregation), and the monomers were agitated at 200 rpm for 4 days, in contrast to our 600 rpm for 7 days incubation. Given the sensitivity of fibril conformations to small modifications in aggregation protocols, we cannot be sure that these fibrils are indeed identical to ours.

Note also that the authors sonicated their fibrils prior to disaggregation. By working with on average much shorter fibrils than in this study, depolymerisation rates would have been higher (due to the presence of more fibril ends) which could obfuscate putative fragmentation events.

Irrespective of these differences, our sucrose density gradient methodology allows the direct isolation of fibril fragments /oligomers and thus we believe we convincingly show that such species are created during chaperone mediated disaggregation.

4. The authors state that the initial aggregation rate of α -syn monomers seeded with the F65 low-speed centrifugation fraction is reduced by 25%, indicating that 75% of the starting material has been processed into smaller fibrils or oligomers in the soluble fraction. However, this consistency seems unlikely because, after the disaggregation reaction, in addition to 25% of the fibrils being removed by centrifugation, some fibrils should have been converted into monomers with no seeding ability. Therefore, the initial seeding rate of the low-speed centrifugation fraction should be lower than 75%, not exactly 75%.

We agree with the reviewer that in the case of the F65 polymorph, the 75% seeding capacity of the low-speed centrifugation supernatant observed is at the upper limit of what one might expect for a disaggregation pathways solely based on depolymerization. Intuitively, with roughly 50% monomers released, one might expect a lower degree of seeding. However, given that we used non-sonicated (and therefore relatively long) fibrils, a release of 50% monomer probably leads to most fibrils shrinking to roughly half their original length, which may not at all decrease their seeding potential, as the latter only depends on the number of growth-competent ends. Given that most of the literature suggest that depolymerization is the default pathway, we see the fact that seeding does not exceed this hypothetical threshold of 75% as an absence of evidence in favor of fragmentation, and consistent with the default model. This is in contrast with the FM data, where the seeding capacity clearly exceeds the expected values for a depolymerization only pathway and thus fragmentation is likely to have occurred.

We have reformulated this section of the results to clarify our interpretation of this data.

Referee #2 (Report for Author)

Highly important topic: Thanks to the cryoEM revolution, we now are aware of amyloid polymorphism and of its apparent importance in disease: certain polymorphs appear to associate to certain pathologies.

The key question now is: how does the polymorph affect the effect of amyloids on the cell and tissue? Also, what mechanisms lead to polymorph selection in disease?

The current manuscript by Wentink and colleagues addresses an interesting angle in this area, namely how differential interactions with Hsp70 chaperones (and cochaps) with polymorphs of alphasynuclein leads to different processing of asyn amyloids by the chaperone. This capitalises on previous work of Ronald Melki on the controlled production of various polymorphs and of the Nussbau-Krammer and Bukau labs on the disassembly of amyloids by Hsp70. With Alex Buell thrown in for good (biophysical) measure, quite the cast;-)

Perhaps not surprisingly to a structural biologist (but still important to show), the atomic detail of the polymorphs appear to matter: they affect how and how well Hsp70/DNJB1 can engage the fibrils. In some cases this can lead to efficient clearance, in some cases it actually produces seeds that promote further aggregation (this was already known from propagation studies of the yeast prion sup35 many years ago, it would be good to cite those).

We thank the reviewer for their kind words. We agree that our observations in the human Hsp70 system mirror those of the yeast bi-chaperone system. We now point out to the similarities between our observations and those made for the yeast prion Sup35 on the impact of structural polymorphism on chaperone action in the discussion section.

I am enthusiastic about this paper, but the part on seeding is a little unclear to me (Figures 4 and 5), and I find this finding important

- seeding is done with intact fibrils? The standard in the field is to sonicate fibrils to get maximal seeding efficiency. How does that look on this scale (it should be much stronger)

All seeding experiments are indeed performed with unsonicated material. Sonicating the material would strongly fragment fibrils independent of chaperone action and would thus potentially mask these effects, as it may have in the past (see our response to minor comment 3 of reviewer 1). We have in the past performed seeding experiments comparing sonicated fibrils to unsonicated fibrils in vitro and in our reporter cell model as shown in the figure below for the XG polymorph, but have not consistently performed these experiments with the fibril batches used in this study and are thus not included in the manuscript. The results are as expected that both in vitro and in cellulo seeding is enhanced with sonicated fibrils.

Fig: A) Aggregation of 100 μ M alpha-synuclein monomers monitored by ThT fluorescence in the absence (grey) or presence of unsonicated (blue) or sonicated (light blue) fibrils. B) Quantification of the percentage of cells with fluorescent foci for HEK239T reporter cells treated with buffer, unsonicated (blue) or sonicated (light blue) fibrils.

- These curves don't look like typical seeding curves: seeding is usually defined as the disappearance of a lag phase and as a quantification, you would show the lag phase in the seeded and unseeded conditions. Here there is no lag phase to begin with, so it looks more like an acceleration of the aggregation reaction, not as the bypassing of the rate-limiting nucleation step

Bypassing the rate-limiting nucleation step by providing sufficient seeds would indeed reduce the aggregation reaction to solely fibril elongation (and potential secondary processes) and thus we believe that the shortening (or complete abolishment) of the lag-phase or the initial aggregation rate are two manifestations of the same phenomenon: "seeding".

The control curve (monomers only, Fig. 4A) demonstrates that there is no spontaneous nucleation of amyloid fibrils on the timescale of these experiments, and thus that in all three sampled seeded conditions (fibrils only, no disaggregation, and disaggregation) the lag-

phase is completely abolished. In this scenario, we thus believe that the initial aggregation rate is a good proxy for the number of “seeds” (growth-competent ends) present in each sample but indeed does not determine if the created seeds are meaningfully different in their ability to template aggregation compared to the original fibrils. Given that we have heterogeneous mixtures of oligomeric species and various (difficult to quantify) amounts of chaperones that co-sediment during the fractionation protocol, we believe assessing the “seeding capacity” per particle is not feasible within our current experimental design.

- It is unclear to me how the 'seeding (normalised)' values are computed here
- The y-axis "Tht signal", specify how it was normalised (and show unnormalised curves in supplement),
- in the quantification of the biosensor microscopy images, the y axis is labelled as 'cells with foci'. Is that fraction or percentage? is the transfection efficiency somehow corrected for or not?

We now provide additional information regarding the normalisation in the figure legends. Note that data in Fig.4 panels A, E and G was not normalized, the axis corresponding to fluorescence intensity in arbitrary units. We have modified the figure to now show the raw values from the experiment. The initial aggregation rates (the slopes at early timepoints) have been normalized to those of the fibril only samples of the same experiments to ease comparison across multiple experiments.

In Fig.5 we counted the percentage cells with foci of the total, and then normalized this to the “no disaggregation” condition within each independent experiment, to express the relative (fold) change across conditions. This indirectly corrects for potential differences in transfection efficiencies across the biological replicates (but assumes transfection efficiencies are the same across conditions within a single experiment).

Is there any high resolution cryoEM data on these polymorphs in the public domain? In this case they should be shown and an attempt should be made to interpret them in light of these data.

The FM polymorph has been structurally characterized by cryoEM, identifying two conformations with identical core structures, but differing in the contacts between filaments (Guerrero-Ferreira et al, 2019). A recent study (Monistrol et al, 2025) now also presents cryoEM data for the XG polymorph, resolving 3 distinct structures, one showing a great degree of similarity to conformation 2a of the FM fibrils, and 2 novel conformations. Note that these are structures of DNAJB1 bound amyloid fibrils. A key difference is the lack of incorporation of N-terminal sequences in one of the conformations found in the XG sample suggesting this region is generally more accessible under these aggregation conditions.

A putative trend therefore is a that a higher degree of disorder in the N-terminus (or more generally a smaller core sequence) results in fibrils more sensitive to disaggregation. However, these observations do not readily explain the differences observed in DNAJB1 affinities, which engages the C-terminal tail that is unstructured in all polymorphs and correlates most strongly with disaggregation activity.

We now describe this hypothesized structural basis for chaperone sensitivity in the discussion section.

I think the main limitation of the study is by focussing on 1 particular Hsp70/DNAJ, the authors may be overestimating the stability of certain fibrils in the face of a chaperone attack: perhaps in cellular conditions, other diversity of the chaperones deals with this issue and in the end all polymorphs are destroyed. I think this should be clearly included as a caveat.

This point is well taken, and we have added this to the discussion.

- the time scales of disaggregation and seeding appear to be quite different, it would be interesting to discuss

This is indeed an interesting point of discussion. The rapid rate of aggregation would suggest that re-aggregation could directly compete with the relatively slow disaggregation process, when there is an abundance of free alpha-synuclein monomers, as might be the case within the cellular environment. We have added this consideration to the discussion section.

Referee #3 (Report for Author)

This work explores the hypothesis that the structure (polymorph) of synuclein fibrils might alter their susceptibility to disaggregation by the Hsp70-mediated system. The authors create recombinant synuclein fibrils under different buffer conditions (and use a C-terminal truncation) to alter the polymorph, using partial proteolysis, negative stain TEM and denaturation curves to document the differences between these structures. Then, they treat with the Hsp70 system (either w/ or w/o ATP), using fluorescence labelling, sucrose gradients and ThT assays to quantify impacts on fibril integrity and the relative size of the products. Finally, they use a biosensor assay in cells to measure relative seeding capacity of these products. The work is carefully performed and clearly written. The methods are key to judging this work and they are, in large part, clearly stated. The statistical methods are also clearly described.

From the experiments, the authors show that multiple factors, including intrinsic fibril stability and overall architecture are indeed important in determining whether the Hsp70 system will dismantle synuclein fibrils to low molecules mass (e.g. monomer) structures. For example, the most stable fibril (Ri) is largely impervious to the Hsp70 system. This work adds important details to our understanding of the human disaggregase - with the major strength being the conclusion that fibril conformation impacts disaggregase efficiency. This reviewer only had a couple of points for consideration, largely focused on enhancing clarity of the presentation.

1. While the authors focus on differences in fibril structure, their conclusions formally require that the total amount or length of the fibrils is relatively uniform. For example, under extreme conditions in which comparison are made between samples that contain sparse fibrils vs. samples with numerous fibrils - then one might be "tricked" into thinking that the resulting differences in disaggregase activity are due to the relative structures of the fibrils when they are actually due to molar ratio. While it is challenging to convincingly document fibril concentration, some additional experimental details would be helpful. The methods indicate

that the fibril concentrations are based on monomer concentration. So, did 100 percent of the monomer progress into the fibril form? This efficacy value could be quantified using the sucrose gradient? Or were the fibrils first purified away from unreacted monomer?

The free monomer fractions after aggregation can be seen in Fig.1B. In these initial experiments we avoided centrifuging the fibrils, which could lead to their higher order assembly into clusters and thus result in potential false negative observations. We find that for the different polymorphs, the soluble fraction (centrifugation at 3,600 g, so this supernatant may include shorter fibril fragments) varies, but corresponds to a maximum of 10% of the total alpha-synuclein. We thus believe that any differences caused by this would fall within the experimental noise.

For the sucrose density gradient experiments however, we used fluorescently labelled fibrils where monomers were removed alongside unreacted dyes by centrifugation. In these experiments, the starting material is thus fully fibrillar. Any monomers observed in the non-disaggregated samples (1-5%) correspond to monomers released as the samples re-equilibrate over 16 hours.

We can indeed not rule out that differences in fibril length contribute to the observed differences in disaggregation efficiencies. Anecdotally we have not observed dramatic differences in fibrils lengths after aggregation but have not systematically quantified this. In previous work however, the FM, Ri, F65 and F91 polymorphs were all found to range between 1.1-1.9 μm on average upon aggregation (Makky et al., Sci Report, 2016), with the efficiently disaggregated polymorph F65 on the high end of this range. These observations would suggest that fibril length is not a major contributing factor to the observed differences in disaggregation efficiency.

We have now addressed this lack of further purification of the fibrils after aggregation in the methods section for clarity.

2. It is not entirely clear that these experiments can fully differentiate between the steps of fibril fragmentation vs monomer extrusion. The experiments in Fig 3 are compelling to suggest that fibrils proceed to oligomers and then monomers (for FM), but there does not appear to be full conservation of mass (fluorescence) in the plots. Under those conditions, a small amount of (slow) monomer release directly from fibril could still be possible. There might be ways around this (e.g. lots of kinetic TEM, single particle tracking microscopy, plus separating the bands on gels), but ultimately it might be best to soften the language around the relative contributions of fibril fragmentation and monomer extrusion. To be clear, this reviewer thinks that the authors conclusion is likely accurate, but the experiments are not entirely convincing.

We indeed do not intend to categorically claim that polymorphs are disaggregated either exclusively by depolymerisation, or fragmentation, but rather that fragments accumulate to various levels for the different polymorphs, implying differences in the relative contribution or rates of depolymerization and fragmentation to the disaggregation reaction. We have softened the language around the specifics of the preferred disaggregation pathways throughout the manuscript. We do not think this undermines the key message around the potential risk of seed formation during disaggregation.

Minor:

1. It would be useful to include the volumes for the ThT assays, along with other details such as the microtiter plate type.

We have edited the materials and methods to supplement the missing information.

2. The buffer conditions for the synuclein fibril preparation are really important for this work and it might be worth putting the conditions in a Table with the citations (which could be easier to read/compare) than in text.

We have added information to Fig.1 with the specific aggregation conditions of the different polymorphs to more easily find back this information.

Dear Dr. Wentink,

Thank you for submitting a revised version of your manuscript. Your study has now been seen by all original referees, who find that their previous concerns have been addressed and now recommend publication of the manuscript. There remain only a few mainly editorial points that have to be addressed before I can extend formal acceptance of the manuscript:

- Please double-check to make sure to all relevant funding information in the manuscript is also entered into our submission system. (missing grant number in ms: 17054 for Alzheimer Forschung Initiative (AFI); missing in eJP: France Parkinson and EraPerMed DEEPEN-iRBD project (ANR-22-PERM-0006))
- Please place the keywords below the abstract
- As we are switching from a free-text author contribution statement towards a more formal statement based on Contributor Role Taxonomy (CRediT) terms, please remove the present Author Contribution section and instead specify each author's contribution(s) directly in the Author Information page of our submission system during upload of the final manuscript. See <https://casrai.org/credit/> for more information.
- APPENDIX 1 FILE WITH ToC: title page should contain "Appendix for + ms title" and ToC with the page numbers for the listed items; Appendix figures don't need to be uplidd as individual files, only in Appendix PDF as they are, so I saved them as Related Manuscript Files
- Please save the source data files need to be saved in a scheme one figure/folder and then uploaded as .zip files. E.g. all the Source data files for figure 1 need to be saved in a single folder and this needs to be zipped and then uploaded as "SD figure 1.zip" file. For EV and/or appendix figures, ZIP together all source data. Completed SD checklist should be uploaded as Related Manuscript File.
- Please provide suggestions for a short 'blurb' text prefacing and summing up the conceptual aspect of the study in two sentences (max. 250 characters), followed by 3-5 one-sentence 'bullet points' with brief factual statements of key results of the paper; they will form the basis of an editor-written 'Synopsis' accompanying the online version of the article. Please also provide an altered synopsis image, making sure that the aspect ratio conforms to our website's format - it should be exactly 550 pixels wide and between 300-600 pixels high.
- Please provide the specific URL for BMRB dataset 52999 in the data availability statement.
- Please rename the heading "Materials and Methods" to "Methods".
- Figure Legends (main + EV):
 1. Please note that the exact p values are not provided in the legends of figures 5C, EV3 C, EV4 B.
 2. Please note that information related to n is missing in the legends of figures EV2 C, D; EV3 B, C; EV4 B, EV5 B
 3. Please note that the error bars are not defined in the legends of figures EV1 C; EV2 D, EV5 B"
 4. Please note that the scale bar needs to be defined for figures EV3 A, EV4 A

With best regards,

Cornelius Schneider

Cornelius Schneider, PhD
Editor | The EMBO Journal
c.schneider@embojournal.org

Use the link below to submit your revision:

Referee #1:

The authors have appropriately addressed the issues raised earlier.

I have one more question. The authors have demonstrated that different polymorphs affect the Hsc70-mediated disaggregation pathway and efficiency. Could the high-resolution structures of these α -synuclein fibrils provide further mechanistic insights into this process ?

Additionally, some terminology in the manuscript should be standardized-for example, the inconsistent use of "fibril," "fiber," and "fibre."

Referee #3:

The authors have nicely addressed my concerns. In particular, the description of the experimental methods are more detailed.

All editorial and formatting issues were resolved by the authors.

Dear Dr. Wentink,

I am pleased to inform you that your manuscript has been accepted for publication in the EMBO Journal.

Yours sincerely,

Cornelius Schneider, PhD
Editor
The EMBO Journal
c.schneider@embojournal.org
